# NXT2 is a key component of the RNA nuclear export factor complex in the human testis and essential for spermatogenesis

Ann-Kristin Dicke [1], Ammar Ahmedani[2], Lin Ma [2], Leonie Herrmann [1], Godfried W. van der Heijden [3], Sophie A. Koser [1], Claudia Krallmann[4], Oguzhan Kalyon [5], Miguel J. Xavier [5], Joris A. Veltman [5,6], Sabine Kliesch [4], Nina Neuhaus [4], Noora Kotaja [2], Frank Tüttelmann [1,7] & Birgit Stallmeyer [1,7] ✉

In eukaryotes, the nucleocytoplasmic export of bulk poly(A)⁺-mRNAs through the nuclear pore complex is mediated by the ubiquitously expressed NXT1-NXF1 heterodimer. In humans, *NXT1* has an X-chromosomal paralog, *NXT2*, which exhibits testis-enriched expression, suggesting a role in spermatogenesis. Here, we report the in vivo interaction of NXT2 with crucial components of the nuclear export machinery, including NXF1, the testis-specific NXF1 paralogs NXF2 and NXF3, and nuclear pore complex proteins. Binding to NXF2 and NXF3 is mediated by the NTF2-like domain of NXT2. By identifying infertile men with loss-of-function variants in *NXT2* and *NXF3*, we link the impaired NXT2-NXF activity to disturbed germ cell development. The predominant absence of germ cells in men with NXT2 deficiency indicates its critical function already during fetal or first steps of germ cell development. In contrast, loss of NXF3 affects later stages of spermatogenesis, resulting in quantitatively and qualitatively impaired sperm production.

In eukaryotic cells, the nuclear envelope is an essential barrier with multiple functions. It separates the nucleus, where precursor messenger RNA (pre-mRNA) is transcribed and processed into mature mRNA, from the cytoplasm, where the mRNA is translated into protein. This separation allows quality control by eliminating non-functional RNAs in the nucleus before they enter the translation machinery and is, thus, involved in the regulation of gene expression[1]. In addition, the nuclear envelope prevents macromolecules from translocating freely between the two compartments and has an important structural role in sheltering the genome[2]. To regulate the nucleocytoplasmic trafficking of macromolecules, such as proteins and RNA, massive (~120 MDa)

nuclear pore complexes (NPCs), consisting of multiple copies of about 30 different nuclear pore proteins (nucleoporins/NUPs), are embedded in the nuclear envelope, forming specialized channels[3]. Transcripts that passed nuclear quality control are bound to ribonucleoproteins (RNPs), forming compact globules that are recognized and coated by transcription export (TREX) complexes, which license the handoff to the nuclear export factor heterodimer, composed of NXT1 (also known as p15/p15-1) and NXF1 (also known as TAP)[4,5].

NXF1 is a modular protein consisting of five highly conserved domains[6]: a nuclear localization signal, a non-canonical RNA-

[1]Institute of Reproductive Genetics, Centre of Medical Genetics, University of Münster, Münster, Germany. [2]Institute of Biomedicine, Integrative Physiology and Pharmacology Unit, University of Turku, Turku, Finland. [3]Division of Reproductive Medicine, Department of Obstetrics and Gynecology, Radboudumc, Nijmegen, Netherlands. [4]Centre of Reproductive Medicine and Andrology, Department of Clinical and Surgical Andrology, University Hospital Münster, Münster, Germany. [5]Biosciences Institute, Faculty of Medical Sciences, Newcastle University, Newcastle-upon-Tyne, UK. [6]Institute of Genetics and Cancer, College of Medicine and Veterinary Medicine, University of Edinburgh, Edinburgh, UK. [7]These authors contributed equally: Frank Tüttelmann, Birgit Stallmeyer. ✉e-mail: birgit.stallmeyer@ukmuenster.de

recognition motif domain (RRMD), a stretch of four leucine-rich repeats (LRR), a nuclear transport factor 2 (NTF2)-like domain, and a C-terminal nuclear pore complex (NPC) binding domain. The RRMD and the LRR both belong to the minimal RNA binding domain, that was shown to bind to the constitutive transport element (CTE) of simian type D retrovirus RNA[7]. The NTF2-like domain mediates binding to NXT1[8] and is also critical for cargo mRNA binding[9] while the C-terminal NPC-binding domain establishes the interaction with NUPs[8,10]. NXT1 consists almost exclusively of the NTF2-like domain and functions as a critical cofactor, enhancing nuclear pore complex binding of NXF1[10].

NXF1 is highly evolutionarily conserved and its essential role in the nuclear export of polyadenylated mRNAs is well established from yeast to mammals[11]. In humans, NXT1 and NXF1 are ubiquitously expressed and the NXT1-NXF1 export pathway is involved in bulk mRNA export in diverse tissues[12]. Interestingly, humans have X-chromosomal paralogs of *NXF1*, known as *NXF2*, *NXF3*, *NXF4*, and *NXF5*. Among these, *NXF2* and *NXF3* exhibit a testis-specific expression profile[13]. Furthermore, the human genome also encodes an X-chromosomal *NXT1* paralog, *NXT2*, which, similar to *NXF2* and *NXF3*, also shows a testis-enriched expression profile[14]. The encoded protein shares ~75% amino acid sequence similarity with NXT1[8]. NXT2 binds to NXF proteins in vitro[8], but the cellular function remains unclear.

Interestingly, in eutherians, *NXT2* orthologs have been demonstrated to have different expression profiles in different lineages[14]. In contrast to testis-enriched human *NXT2*, mouse *Nxt2* is ubiquitously expressed[14], suggesting an evolutionarily young and specific role of the human NXT2 in spermatogenesis. Indeed, *Nxt2* has evolved conservatively in mice, implying functional constraints as predominant force[15]. In contrast, adaptive selection, which describes the propagation of advantageous genetic variations through positive selection, has frequently contributed to genetic changes in primate *NXT2*[15]. This quite different evolutionary development of the two NXT2 orthologs indicates that the primate protein has acquired novel substrate- and/or tissue-specific functions that not only differ from the function of NXT1 but also from the function of the murine NXT2[15].

Here, we demonstrate that not only NXF1 but also its testis-specific paralogs NXF2 and NXF3 belong to the human testicular interactome of NXT2 and propose an adapted nuclear RNA export pathway in the human testis with NXT2 as the key component of the RNA nuclear export factor complex. In addition, we provide evidence that an impaired testis-specific NXT2 interactome leads to male infertility. Men with loss-of-function (LoF) variants in *NXT2* show absence of sperm in the ejaculate (azoospermia) and near absence of germ cells in their testes (non-obstructive azoospermia, NOA). In contrast, a man with a LoF in *NXF3* produced very few, and predominantly immotile sperm. Taken together, this indicates that the NXT2 interactome likely plays critical roles during embryonic/fetal germ cell development as well as during spermatogenesis in the adult testis.

## Results

### A nuclear export pathway involving NXT2 in the adult human testis

As *NXT2* shows a testis-enriched expression in humans and has been influenced by adaptive selection in primates[15], we aimed to analyze the interactome of NXT2 in vivo by means of antibody-mediated capture of NXT2 from testis tissue lysates derived from an adult donor with full spermatogenesis (lysate 1, $N = 3$) using a validated antibody, specifically recognizing NXT2 (Supplementary Fig. 1a, Supplementary Table 1). Western blot analyses confirmed successful pulldown of NXT2 (Supplementary Fig. 2a) and in subsequent mass spectrometry analysis (Supplementary Data 1), NXT2 was enriched 22-fold compared to IgG isotype control pulldown (*p*-value: 0.001, Fig. 1a, Supplementary Data 2). Taking into account only those proteins that were identified in three independent pulldown approaches, 43 additional proteins were

at least 5-fold enriched compared to the IgG pulldown control (Supplementary Data 2) and, thus, possibly interact with NXT2 or NXT2 binding partners. STRING interaction analysis of these proteins, considering exclusively experimental-based sources, highlighted three major clusters (Fig. 1b). The core cluster, which includes NXT2, contains three members of the nuclear export factor protein family: NXF1 (10-fold enriched), NXF2 (56-fold enriched), and NXF3 (45-fold enriched) pointing towards a possible function of NXT2 in RNA export through the nuclear pore. This is strengthened by the additional identification of eight nucleoporins in the second cluster. In addition, the mass spectrometry data highlighted two ribosome biogenesis factors, SPATA5 and SPATA5L1 (cluster three). Notably, all three NXF-proteins, the nucleoporins NUP214 and NUP93, SPATA5, and SPAT5L1, were also at least 3-fold enriched in a separate pulldown of NXT2 from a second independent testis lysate (lysate 2, $N = 2$, Supplementary Data 3 and 4) and a pulldown from a third lysate derived from pooled testicular biopsies of three men with obstructive azoospermia (lysate 3, $N = 1$, Supplementary Data 5). Gene Ontology analysis supported a role for NXT2 and its interaction partners in nucleocytoplasmic RNA transport as indicated by the enrichment in terms 'nucleocytoplasmic transport', 'mRNA transport' and 'nuclear export' and 'RNA localization' (Supplementary Fig. 3a). In line, the main cellular components associated with the NXT2 interactome were the 'nuclear pore' and the 'nuclear export factor complex' (Supplementary Fig. 3b).

In contrast to *NXF1*, which is located on chromosome 11 and is ubiquitously expressed, *NXF2* and *NXF3* are located on the X chromosome and are specifically expressed in the testis (Fig. 1d), arguing for a testis-specific RNA export pathway, functionally analogous to the NXT1-NXF1-mediated ubiquitous bulk mRNA export pathway, but involving testis-enriched NXT2 as the key component of the RNA nuclear export factor complex and, in addition to NXF1, possibly also testis-specific NXF2 and NXF3 (Fig. 1c).

To corroborate the proposed direct interaction of NXT2 with NXF2 and NXF3, we performed co-immunoprecipitation (Co-IP) of HA-tagged NXT2 with FLAG-tagged NXF2 (Fig. 1e) or FLAG-tagged NXF3 (Fig. 1f) overexpressed in HEK293T cells. NXF2 and NFX3 were only detectable in Western blots when NXT2 was co-expressed, demonstrating a specific and direct interaction of both proteins with NXT2 also in vitro.

### NXT2 interacts with NXF2 and NXF3 through NTF2-like domains

The protein sequences of NXF2 and NXF3 show 57% and 44% sequence identity, respectively, with NXF1[8]. All three proteins share a non-canonical RNA recognition motif domain (RRMD) and a nuclear transport factor 2 (NTF2)-like domain. While NXF1 and NXF2 also share four central leucine-rich repeats and a C-terminal nuclear pore complex binding domain, these domains are absent from NXF3. Unique to NXF3 is also an XPO1-dependent nuclear export signal, which is thought to compensate for the loss of the nuclear pore domain[16] (Supplementary Fig. 4). To characterize the interaction of NXT2 with testis-specific NXF2 and NXF3 in detail, we queried which of the two shared domains, mediates the binding to NXT2. Accordingly, we generated FLAG-tagged expression constructs of NXF2 and NXF3, lacking the NTF2-like domains or the RRM domains (Fig. 2a and c), over-expressed them in HEK293T cells (Supplementary Fig. 5a), and performed Co-IP analyses. Binding to NXT2 was abolished when the NTF2-like domain in NXF2 was deleted (Fig. 2b). However, NXF2 lacking the RRM domain was still able to bind to NXT2 (Fig. 2d). In line, deletion of the NTF2-like domain in NXF3 (Fig. 2e; Supplementary Fig. 5b) inhibited binding to NXT2 (Fig. 2f), whereas deletion of the RRM domain (Fig. 2g; Supplementary Fig. 5b) did not affect binding capacity (Fig. 2h). Besides, a C-terminally truncated NXT2, lacking the last six amino acids but containing the NTF2-like domain (Supplementary Fig. 6a) was still able to bind to both NXF2 (Supplementary Fig. 6b) and NXF3 (Supplementary Fig. 6c). These data show that the binding of

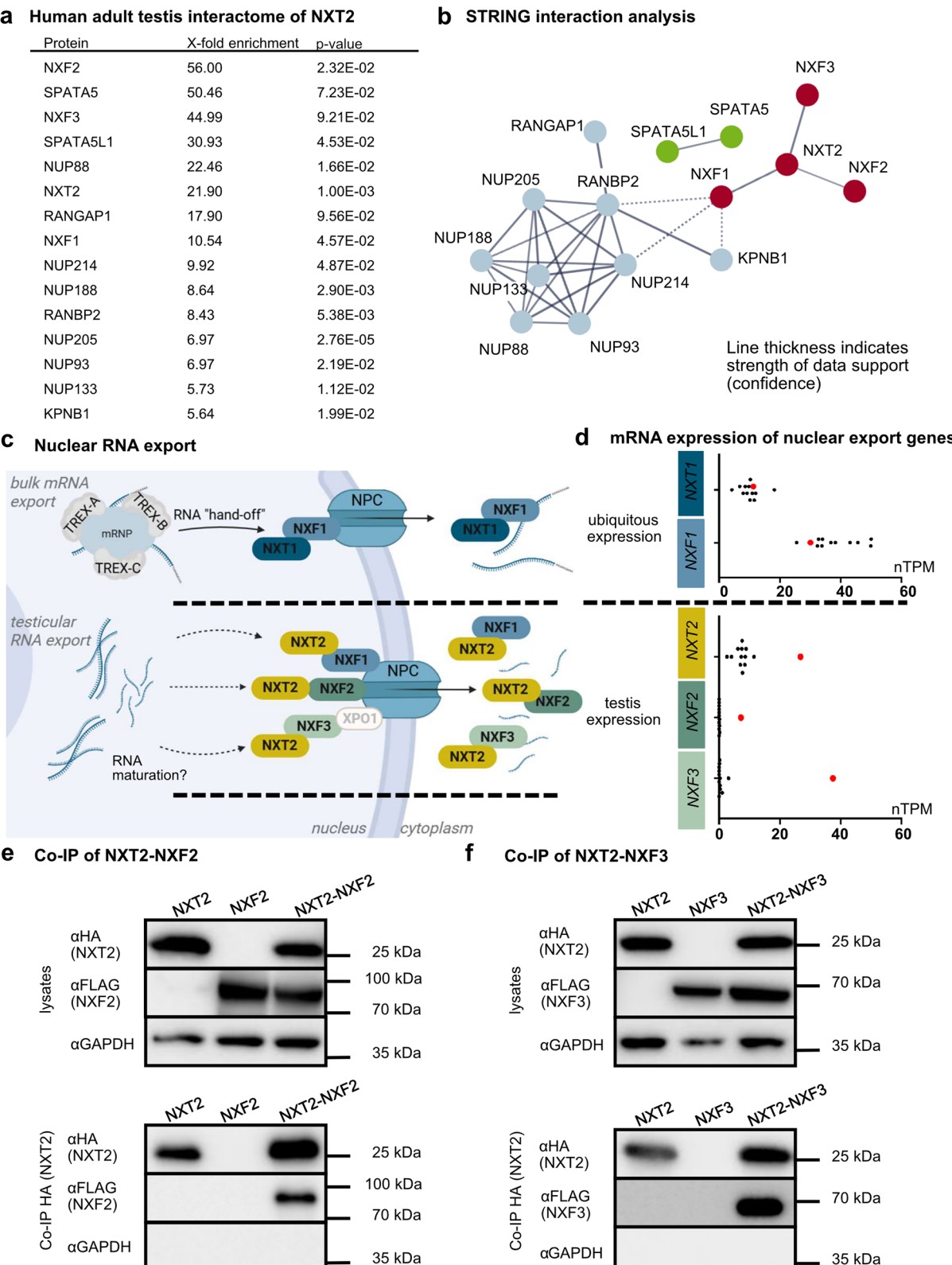

**a** Human adult testis interactome of NXT2

| Protein | X-fold enrichment | p-value |
|---|---|---|
| NXF2 | 56.00 | 2.32E-02 |
| SPATA5 | 50.46 | 7.23E-02 |
| NXF3 | 44.99 | 9.21E-02 |
| SPATA5L1 | 30.93 | 4.53E-02 |
| NUP88 | 22.46 | 1.66E-02 |
| NXT2 | 21.90 | 1.00E-03 |
| RANGAP1 | 17.90 | 9.56E-02 |
| NXF1 | 10.54 | 4.57E-02 |
| NUP214 | 9.92 | 4.87E-02 |
| NUP188 | 8.64 | 2.90E-03 |
| RANBP2 | 8.43 | 5.38E-03 |
| NUP205 | 6.97 | 2.76E-05 |
| NUP93 | 6.97 | 2.19E-02 |
| NUP133 | 5.73 | 1.12E-02 |
| KPNB1 | 5.64 | 1.99E-02 |

**b** STRING interaction analysis

Line thickness indicates strength of data support (confidence)

**c** Nuclear RNA export

**d** mRNA expression of nuclear export genes

**e** Co-IP of NXT2-NXF2

**f** Co-IP of NXT2-NXF3

NXT2 and NXF2/3 depends on the NTF2-like domain of the respective interaction partners, while the RRM domain is dispensable in vitro.

## NXT2 is the predominant NXT protein in the human adult testis and part of the NXF3 interactome

NXT1 was absent from the NXT2-NXF interactome. However, according to transcriptomic data of the human protein atlas, *NXT1* is still

expressed in the human adult testis, albeit at lower level than *NXT2* (Fig. 1d). To address the question of whether NXT1 is part of the interactome of one of the testis-specific NXF proteins and could potentially replace the function of NXT2, we performed a pulldown of NXF3 and its binding partners from human adult testis protein lysate (lysate 1, $N = 3$), using a validated NXF3-specific antibody (Supplementary Fig. 1b). Western blot analysis confirmed the successful

**Fig. 1 | The testicular interactome of NXT2 indicates a testis-specific nucleo-cytoplasmic RNA export pathway. a** List of proteins detected in mass spectrometry data of an NXT2 pulldown (*N* = 3) from adult human testis indicating proteins at least 5-fold enriched compared to the IgG control and present in STRING clusters of NXT2 interactome. *P*-values refer to one-sided *t*-test on protein abundance values. Raw data are presented in Supplementary Data 1 and 2. **b** The STRING interaction analysis of at least 5-fold enriched proteins identified three main clusters. The central cluster (red) includes NXT2 and the nuclear export factors NXF1, NXF2, and NXF3. The blue cluster comprises eight nucleoporins including NUP93 and NUP214, and the green cluster includes the ribosome biogenesis factors SPATA5 and SPATA5L1. Only high confidence (≥0.7) associations based on experimental sources are shown. Disconnected nodes are hidden. **c** Proposed models for testis-specific RNA export pathways with NXT2 as the key component of the RNA nuclear export factor complex. At the top, the well-studied ubiquitous nuclear bulk mRNA export pathway involving NXT1 and NXF1 is depicted. In the middle and at the bottom, two possible testis-specific nuclear export pathways are shown involving NXT2 and its testicular binding partners NXF2 and NXF3, respectively. Created in BioRender. Stallmeyer, B. (2025) https://BioRender.com/vbcswtj. **d** Box plots comparing mRNA expression data of human nuclear export factor genes in 12 diverse human tissues (black dots) with the expression in the testis (red dots). Expression data are derived from the Human Protein Atlas (nTPM = normalized transcripts per million; Source data are shown in the Source data file). **e** Western blot analysis of Co-IP of NXT2-HA and NXF2-FLAG demonstrates that NXF2 binds to NXT2 in vitro in lysates derived from NXT2-NXF2 co-transfection in HEK293T cells. **f** Binding of NXT2 to NXF3 was corroborated in NXT2 Co-IP of overexpressed proteins as shown in subsequent Western blot analysis (bottom). Representative Western blots from at least three replicates (*N* = 3) are shown. nTPM: normalized transcripts per million. TREX: Transcription export complex. Source data are provided as a Source Data file.

pulldown of NXF3 (Supplementary Fig. 2b), and subsequent mass spectrometry analysis (Supplementary Data 6) showed a 107-fold enrichment of NXF3 compared to IgG pulldown controls (*p*-value: 0.07; Supplementary Data 7). Interestingly, no NXT1 was detected in the mass spectrometry data of any of the three independent NXF3 pulldown approaches, whereas NXT2 was not only called in all three samples, but also enriched 8-fold (*p*-value: 0.11), compared to IgG control pulldowns (Supplementary Data 7), indicating that NXT2 is the predominant NXT protein interacting with NXF3 in the human adult testis. The absence of NXT1 from NXF3 pulldown was confirmed in an independent pulldown from testis lysate 2 (*N* = 1, Supplementary Data 8).

Since NXT1 was not detectable in the NXF3 pulldown, we aimed to address the question of whether NXT1 is at all capable of binding to testis-specific NXF2 and NXF3. In Co-IP analysis of full-length HA-tagged NXT1 with FLAG-tagged NXF2 and NXF3, respectively, NXF2 and NFX3 were only detectable when NXT1 was co-expressed, demonstrating that both proteins can bind to NXT1 in vitro (Supplementary Fig. 7) and that the absence of NXT1 from the NXF3 pulldown might be a consequence of low NXT1 protein levels in the human adult testis. Accordingly, we attempted to pull down NXT1 from human adult testis (lysate 1, *N* = 3) using an NXT1-specific monoclonal antibody (Supplementary Fig. 1c). In subsequent Western blot analysis, only a faint NXT1-specific signal was detectable in three independent approaches, even when the amount of antibody used for pulldown was doubled compared to the NXT2 and NXF3 pulldown approaches (Supplementary Fig. 2c). Consistent with this, no NXT1 was identified in subsequent mass spectrometry in any of the samples (Supplementary Data 9). In addition, no NXT1 specific staining was detectable in sections of testicular biopsies with full spermatogenesis in immunohistochemical analyses (Supplementary Fig. 8). In summary, these data support that NXT2 is the key binding partner of the NXF proteins in the human adult testis.

## Deleterious variants in *NXT2* are associated with azoospermia and loss of germ cells

Since NXT2 has a testis-enriched expression profile and may have evolved testis-specific functions, we next addressed the question of whether impaired NXT2 function in the human testis impairs sperm production and causes male infertility. A first indication comes from population genetic data: NXT2's LoF observed/expected (o/e) fraction is zero with a LoF o/e upper bound fraction (LOEUF) of 0.51 (gnomAD, v2.1.1, not yet available for v.4.1, Supplementary Table 2). Such low o/e fractions are rare and specify intolerance to LoF variants, supporting an important biological relevance of the encoded protein and selective pressure on genetic variants affecting protein function. Indeed, *TEX11* is one such X-chromosomal gene with an o/e fraction of zero and one of the best established male infertility genes[17,18]. We, therefore, screened for potentially deleterious genetic variants in *NXT2* in exome/

genome sequencing data of >2700 well-characterized infertile men from the Male Reproductive Genomics (MERGE) cohort. Focusing on rare (minor allele frequency [MAF] ≤0.001, gnomAD v2.1.1) LoF or high impact missense variants (with CADD score ≥10) and copy number variations in *NXT2*, we identified two infertile men. Subject M3065 showed a hemizygous duplication of a T at position 354 of the open reading frame of *NXT2* isoform 1 (ENST00000372103.1, NM_018698.5; c.354dup), resulting in a premature stop codon at aspartic acid 119 [p.(Asp119*)] (Fig. 3a, Supplementary Table 2), and M2004 the single nucleotide substitution c.268G>T resulting in the substitution of an alanine to serine at position 90 in the NTF2-like domain of the NXT2 protein sequence [p.(Ala90Ser)] (Fig. 3a, Supplementary Table 2). In addition, a further LoF variant in *NXT2* was identified in exome data of a second cohort of 667 infertile men from Nijmegen/Newcastle[19]. The respective subject RU00584 was positive for a large deletion encompassing the entire *NXT2* gene (Fig. 3a, Supplementary Table 2). This variant had been listed in the supplemental data of a recent publication without presenting clinical or functional data[19]. Andrological evaluation in all three men concordantly described azoospermia and elevated FSH levels (Supplementary Table 3), indicating impaired spermatogenesis.

To investigate the impact of disturbed NXT2 function on testicular germ cell development, we analyzed the subjects' clinical and testicular phenotypes in detail, as well as the cellular expression pattern of *NXT2* in the human testis. Periodic acid-Schiff (PAS) staining of testicular sections from all three cases, derived from testicular biopsies, revealed a common phenotype characterized by seminiferous tubules predominantly lacking germ cells (Sertoli cell-only, SCO, HP:0034299, Fig. 3b). To reinforce these histological findings, we performed immunohistochemical staining for the germ cell/spermatogonial stem cell marker MAGEA4. Interestingly, only M3065 had a few focal seminiferous tubules with spermatogonia and spermatocytes (Fig. 3c), but no sperm could be retrieved by testicular sperm extraction (TESE). In contrast, MAGEA4 staining was negative in M2004 and RU00584, although a few morphologically abnormal sperm were detected in a TESE-derived testicular cell suspension in RU00584. Due to their abnormal appearance, the sperm were deemed unsuitable for ICSI (Supplementary Table 3).

Published scRNA-seq datasets[20,21] of the human fetal male germ cells and adult testis depict a strong expression of *NXT2* not only in adult but also in fetal germ cells (Fig. 3d), supporting a function of NXT2 already at fetal/embryonic stages of germ cell development. In addition to germ cells, in adult testicular tissue, *NXT2* was also expressed in somatic Sertoli cells[21–23], indicating not only a developmental, but also a cell type-specific function of NXT2 in testicular tissue (Fig. 3d).

To address the genetic and functional impact of the identified variants in *NXT2*, we performed co-segregation analyses within the families and in-depth functional characterization of the variants at the

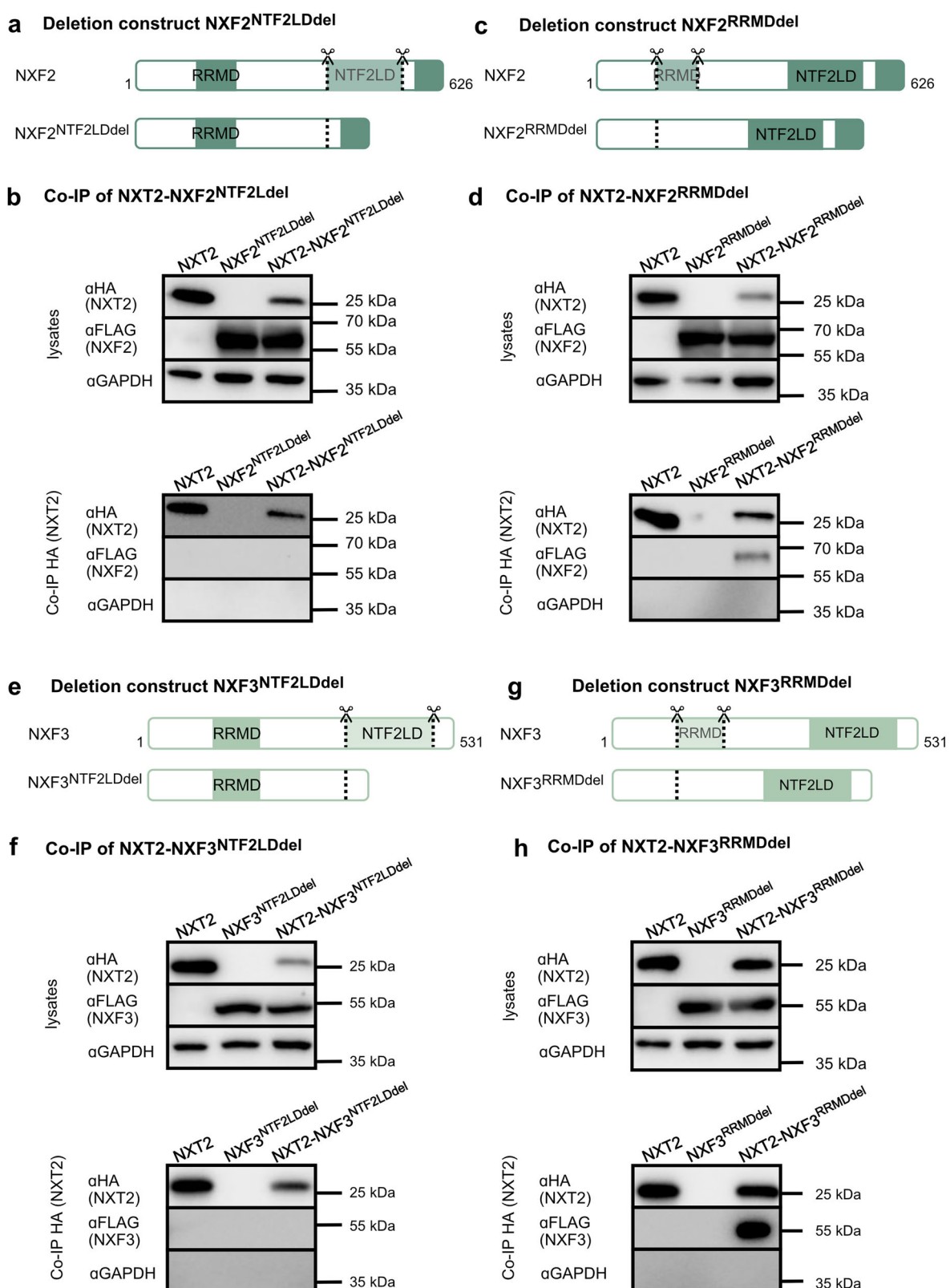

**a** Deletion construct NXF2^NTF2LDdel

**b** Co-IP of NXT2-NXF2^NTF2Ldel

**c** Deletion construct NXF2^RRMDdel

**d** Co-IP of NXT2-NXF2^RRMDdel

**e** Deletion construct NXF3^NTF2LDdel

**f** Co-IP of NXT2-NXF3^NTF2LDdel

**g** Deletion construct NXF3^RRMDdel

**h** Co-IP of NXT2-NXF3^RRMDdel

protein level. M3065 shared the *NXT2* LoF variant c.354dup p.(Asp119*) with two further infertile brothers with azoospermia (M3065B1/B3). The variant was inherited from the heterozygous mother and was absent in both the fertile brother and the father, thus co-segregating with azoospermia/infertility (Fig. 4a). Other genetic causes were excluded by filtering for shared rare (MAF ≤0.01), coding variants in the exome data of all affected family members that were absent in

unaffected male family members. In this shared variant analysis, additional rare variants, predicted to result in protein sequence alterations and additionally co-segregating with the phenotype, were identified exclusively in genes that were expressed in other tissues or ubiquitously expressed (Supplementary Table 4).

The c.354dup variant introduces a premature stop codon at position 119 of the *NXT2* isoform 1 open reading frame, likely resulting

**Fig. 2 | NXT2 binds to NXF2 and NXF3 by their NTF2-like domains. a** Schematic depiction of deletion construct of NXF2-FLAG lacking the NTF2-like domain (NXF2$^{NTF2LDdel}$). **b** Western blot analyses demonstrating specific expression of NXT2 and NXF2$^{NTF2LDdel}$ in protein lysates isolated from transfected HEK293T cells (top). In Co-IP analysis, NXF2$^{NTF2LDdel}$ is not able to bind to NXT2 (bottom). **c** Schematic depiction of deletion construct of NXF2 lacking the RNA recognition motif domain (NXF2$^{RRMDdel}$). **d** NXF2$^{RRMDdel}$ still binds to NXT2 (cell lysates derived from transfected HEK293T cells at the top, Co-IP at the bottom). **e** Schematic depiction of deletion constructs of NXF3-FLAG lacking the NTF2-like domain (NXF3$^{NTF2LDdel}$). **f** In Co-IP analysis, NXT2-NXF3 heterodimerization is dependent on the NTF2-like domain in NXF3. **g** Schematic depiction of deletion construct of NXF3 lacking the RNA recognition motif domain. **h.** In Co-IP analysis of NXT2 and NXF3$^{RRMDdel}$, binding of both proteins is still possible, indicating that the RRM domain is dispensable for dimerization. Representative Western blots of at least three replicates are shown ($N = 3$). Source data are provided as a Source Data file. RRMD: RNA-recognition motif domain, NTF2LD: NTF2-like domain.

in the degradation of the mutant mRNA by nonsense mediated decay (NMD). If, however, the transcript escapes NMD, the truncated protein would lack 78 C-terminal amino acids and thus most of the NTF2-like domain (Fig. 4b). Overexpression of HA-tagged NXT2 c.354 dup in HEK293T cells resulted in the complete absence of the truncated protein according to Western blot analysis (Fig. 4c), arguing against the expression of a stable truncated protein. To corroborate the suspected absence of NXT2 in the subject's testicular tissue, we performed immunohistochemical staining for NXT2 using an antibody that is directed against an NXT2 protein region that would be present in the truncated protein (Supplementary Table 1). Indeed, no NXT2-specific staining could be detected in the subject's Sertoli cells, which were clearly assignable by positive staining for the Sertoli cell marker SOX9. In contrast, a tissue with complete spermatogenesis displayed positive staining for NXT2 in spermatogonia and Sertoli cells (Fig. 4d).

RU00584 was previously described as carrying a deletion of *NXT2* identified by exome sequencing[15], but this variant had not been prioritized for follow-up at that time. Analysis of subsequently produced genome sequencing data indicated that the subject presents a 42 kb large deletion on the X chromosome encompassing the entire *NXT2* gene. The region does not contain additional protein or long non-coding RNA encoding genes, but it covers the distal enhancer of the adjacent *KCNE5* gene. *KCNE5* encodes a regulatory subunit of a cardiac potassium channel, and it is not expressed in the human testis. Therefore, the deletion of this regulatory region is unlikely to be responsible for the patient's infertility. Interestingly, the deletion occurred de novo, as it was not detected in the subject's parents, providing further genetic evidence that the variant is pathogenic (Fig. 5a). Fittingly, no NXT2-specific staining could be detected in Sertoli cells, identified by SOX9-positive staining, confirming the expected absence of NXT2 in vivo (Fig. 5b).

The third *NXT2* variant identified in M2004, c.268G>T, affects the 5'-end nucleotide of *NXT2* exon four (Supplementary Fig. 9a). As this variant might affect splicing, we performed a minigene assay to analyze the effect on the recognition of the respective splice site in vitro. In contrast to the wildtype, the mutant *NXT2* minigene revealed two distinct splice products. In addition to the clearly visible normally spliced transcript, a faint signal was detected corresponding to an aberrantly spliced transcript lacking exon four. Skipping of this exon would result in a shift of the open reading frame and subsequent insertion of a premature stop codon [c.268G>T, r.268_412del p.(Ala90Serfs*13)] (Supplementary Fig. 9b). The predicted amino acid substitution resulting from c.268G>T affects the alanine residue at position 90, which is highly conserved in orthologous proteins (Supplementary Fig. 9c) and positioned in a central protein domain, as indicated by the AlphaFold2 model (Supplementary Fig. 9d). Because the residue is located in the NTF2-like domain, which is crucial for binding to NXF2 and NXF3, we next analyzed whether the amino acid substitution affects protein stability or binding to NXF2 or NXF3. When overexpressing mutant NXT2 in HEK293T cells, the protein expression level was comparable to the wildtype (Supplementary Fig. 9e) and, in Co-IP, binding of mutant NXT2 to NXF2 (Supplementary Fig. 9f) and NXF3 (Supplementary Fig. 9g) was unaffected. In addition,

NXT2 staining was present in SOX9-positive Sertoli cells in the subject's testis (Supplementary Fig. 9h). Accordingly, no clear functional link could be established between this missense variant and the proband's Sertoli cell-only phenotype.

**An *NXF3* variant leads to oligoasthenoteratozoospermia**

To unravel whether the testis-expressed binding partners of NXT2, the nuclear export factor proteins NXF1, NXF2, and NXF3, are also important for male fertility, we screened the exome/genome data of the MERGE and Nijmegen/Newcastle cohorts for rare, potentially deleterious variants in *NXF1*, *NXF2*, and *NXF3* using the same filtering criteria as outlined above for *NXT2*.

A potentially deleterious variant was detected only in *NXF3*. One man with infertility (M2799) identified in the MERGE cohort was positive for the hemizygous stop-gain variant at position 826 of *NXF3* isoform 1 (ENST00000395065.8, NM_022052.2:c.826G>T), leading to a premature stop codon at glycine 276, p.(Gly276*). With a LoF o/e fraction of 0.33 and a LOEUF of 0.6 (gnomAD, v2.1.1) (Supplementary Table 2) also *NXF3* indicates a reduced tolerance to LoF variants in the general population. The variant is located in exon nine of *NXF3* (Fig. 6a) and would lead to a protein lacking the NTF2-like domain (Fig. 6b) if the mutant transcript escaped NMD. The variant was inherited from the heterozygous mother and was not identified in two fertile brothers of the mother (Fig. 6c). No further high-impact variants in genes expressed in the testis were present in the proband's exome. In Western blot analyses of lysates overexpressing the mutant protein, a ~30 kDa smaller protein (Fig. 6d) compared to the wildtype protein was detected. This truncated protein was unable to bind to NXT2, as demonstrated by Co-IP (Fig. 6e). Thus, even if the mutant transcript would escape NMD, the stop-gain variant would result in an abolished interaction with NXT2.

In contrast to the subjects with *NXT2* variants, who were concordantly azoospermic, M2799 had a strikingly reduced sperm count in the ejaculate (extreme oligozoospermia, <2 million total sperm count, HP:0034815), mostly immotile sperm (≥85%) in two independent semen samples, and, strikingly, 0% normally formed sperm. In particular, 90% of sperm showed diverse head defects, 43% midpiece defects and 78% tail defects, including coiled tails as the most prominent phenotype (Fig. 7a, Supplementary Fig. 10). The striking difference in the phenotypes observed in men with *NXT2* variants compared to the phenotype of M2799 with the *NXF3* variant is in line with the different expression profiles of the two genes. In contrast to *NXT2*, *NXF3* is not expressed in male fetal germ cells, according to published scRNA-seq datasets (Supplementary Fig. 11a). Notably, within the adult testis, *NXF3* mRNA is present in Sertoli cells and spermatids (Supplementary Fig. 11b), indicating a function of the NXT2-NXF3 heterodimer at later stages of spermatogenesis.

To analyze the localization of NXF3 in sperm, we performed immunofluorescence staining using an NXF3-specific antibody (Supplementary Fig. 1c/d; Supplementary Table 1). In sperm from a control donor, NXF3 localized to the midpiece, more specifically to the centriole/connecting piece (Fig. 7b). In contrast, no NXF3-specific staining was detected in M2799's sperm, indicating the absence of

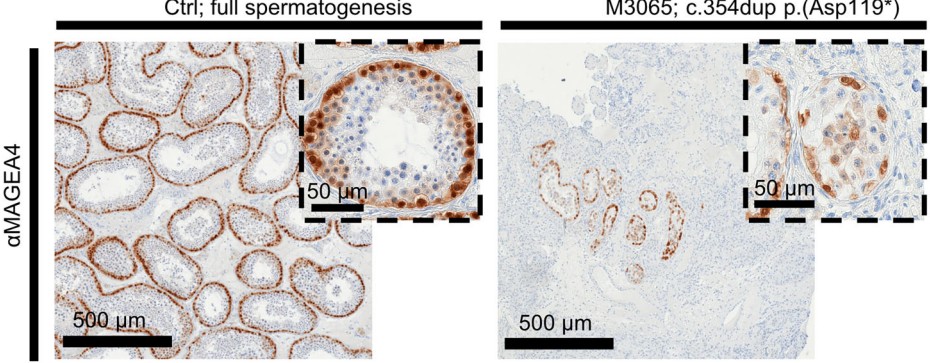

**a   Localization of NXT2 variants**

**b   Testicular phenotypes**

**c   Germ cell detection via MAGEA4 staining**

**d   NXT2 expression**

protein expression in vivo. To further screen for morphological abnormalities in the midpiece we stained the sperm with Mitotracker, staining mitochondria in the midpiece (Fig. 7b). Interestingly, in M2799's sperm, Mitrotracker-specific staining was not detectable, further supporting the observed midpiece defects seen in the Papanicolaou staining or indicating a possible mitochondrial dysfunction. In summary, the man with the *NXF3* LoF variant had quantitatively and qualitatively severely impaired sperm production (oligoasthenoteratozoospermia).

## Discussion

In this study, we demonstrate that testis-enriched NXT2 is indispensable for normal spermatogenesis in humans by identifying two *NXT2* variants that abolish protein expression in infertile men with

**Fig. 3 | Loss-of-function variants in *NXT2* are associated with male infertility due to disturbed germ cell development. a** Schematic depiction of the *NXT2* exon-intron-structure and the NXT2 primary protein structure with the position of NTF2-like domain highlighted. Red dots indicate the localization of the identified *NXT2* variants at the gene and protein level. The missense variant p.(Ala90Ser) identified in M2004 and the LoF variant p.(Asp119*) identified in M3065 are both located within the NTF2-like domain. The deletion identified in RU00584 encompasses the entire *NXT2* gene. **b** Overview of the testicular phenotype of all three men with variants in *NXT2* (PAS staining of testicular sections) compared to a control with full spermatogenesis (*N* = 3). All subjects consistently had a Sertoli cell-

only phenotype, i.e. a complete lack of germ cells, in the vast majority of seminiferous tubules. Sertoli cells are exemplary marked by arrows. **c** Immunohistochemical staining for germ cell marker protein MAGEA4 reveals only sparse seminiferous tubules with few spermatogonia and spermatocytes only in M3065 (*N* = 2). No haploid germ cells were identified. **d.** In fetal male germ cells (left), NXT2 is highly expressed in germ cells according to scRNA-seq data. In the adult testis (right), NXT2 is mainly expressed in spermatogonia but also in Sertoli cells and in all germ cell stages up to spermatids in two different data sets (EC endothelial cells, FLC fetal Leydig cells, LC Leydig cells, SC Sertoli cells, GC germ cells, SPG spermatogonia, SPC spermatocytes, RS round spermatids).

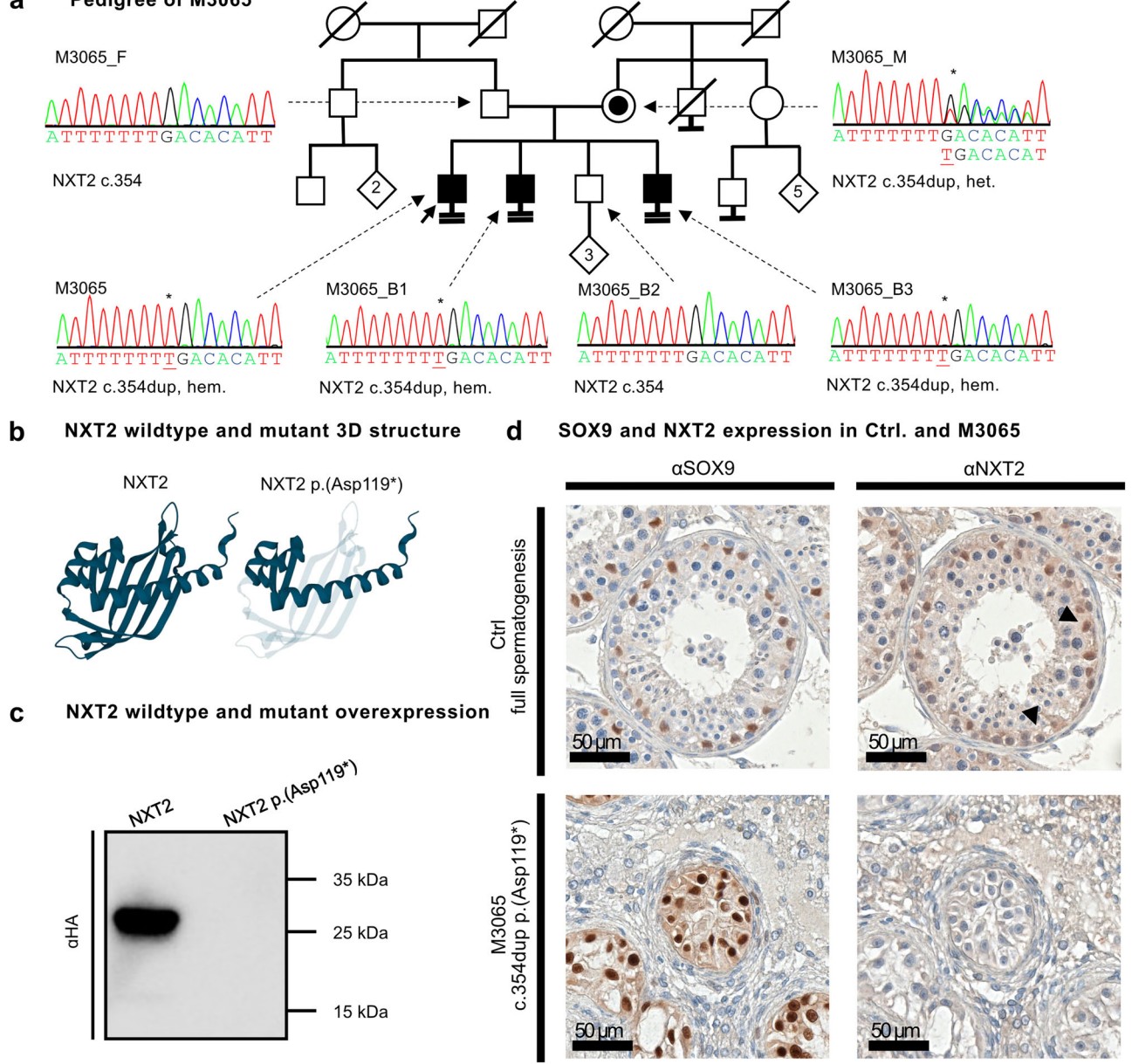

**Fig. 4 | The *NXT2* variant c.354dup p.(Asp119*) co-segregates with azoospermia and leads to NXT2 deficiency. a** Index subject M3065 carries a duplication of a thymine at position 354 of the *NXT2* open reading frame resulting in the direct inclusion of a premature stop codon. The variant is inherited from the mother (heterozygous carrier) and is also present in two brothers with azoospermia. The variant is absent in the only fertile brother. **b** AlphaFold2 prediction of the short isoform (NM_001242617.2) of wildtype NXT2 (left). The inclusion of a premature

stop codon results in the loss of 78 amino acids at the protein C-terminus (transparent, right). **c** In Western blot analysis of protein lysates derived from overexpression of mutant NXT2-HA, no protein was detectable (*N* = 3). Source data are provided as a Source Data file. **d** Sertoli cells show positive SOX9 staining in control's and M3065's testicular tissue. In contrast, NXT2-specific staining, which is mainly visible in Sertoli cells and spermatogonia in the control tissue (indicated by arrows), is absent in the subject (*N* = 3).

## a  CNV Plot of RU00285

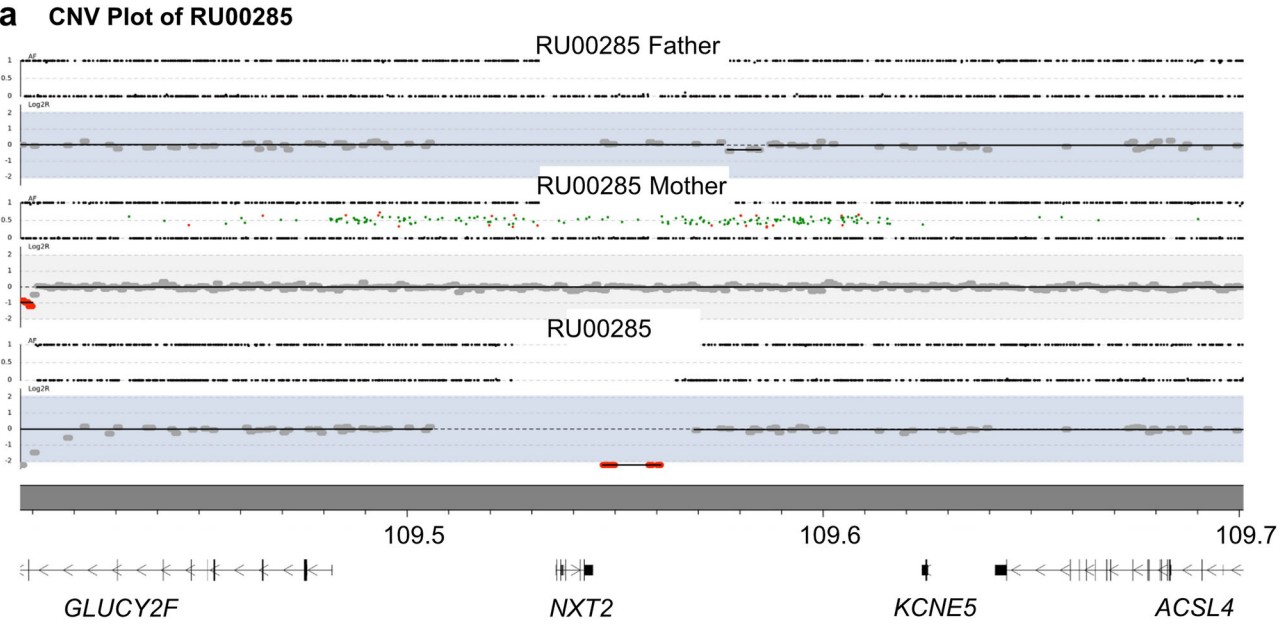

GLUCY2F          NXT2          KCNE5     ACSL4

## b  SOX9 and NXT2 expression in Ctrl. and M3065

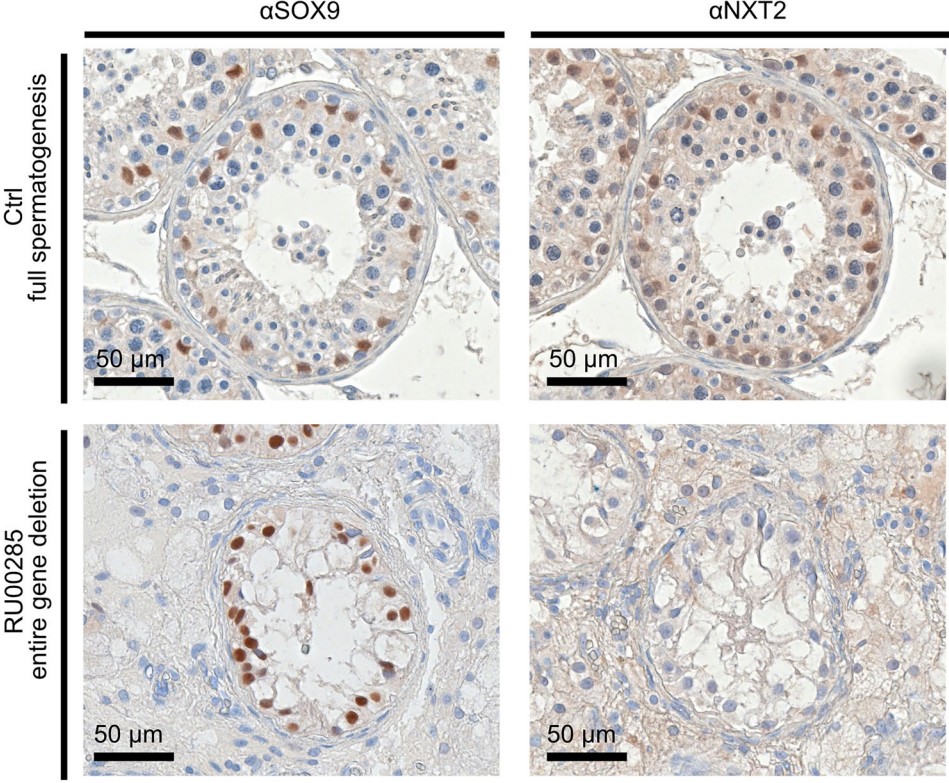

**Fig. 5 | Deletion of the entire *NXT2* gene in RU00584 occurred de novo and results in the absence of NXT2 in vivo. a** Robot (CNV) plot of genome sequencing data illustrating the presence of a 42 kb deletion in RU00584 encompassing the entire *NXT2* gene and surrounding genomic regions. The deletion is absent in both parents, demonstrating de novo occurrence in the subject. **b** In vivo, immunohistochemical staining for the Sertoli cell marker protein SOX9 confirms the presence of Sertoli cells in the control (top) and in RU00584 (bottom). In an immunohistochemical staining for NXT2, in contrast to the control, no NXT2-specific staining was observed in Sertoli cells in RU00584 (*N* = 2).

azoospermia. Both subjects share a testicular phenotype of predominant Sertoli cell-only, *i.e.*, absence of germ cells, which we link to the absence of NXT2 in the testicular tissue. One of the variants cosegregates with the infertility in the family, while for the second variant a de novo occurrence was demonstrated, and both of these observations strengthen the genetic evidence for the association of *NXT2* with the observed phenotype[19]. Indeed, *NXT2* had not yet been associated with any disease and only three heterozygous LoF variants in *NXT2*,

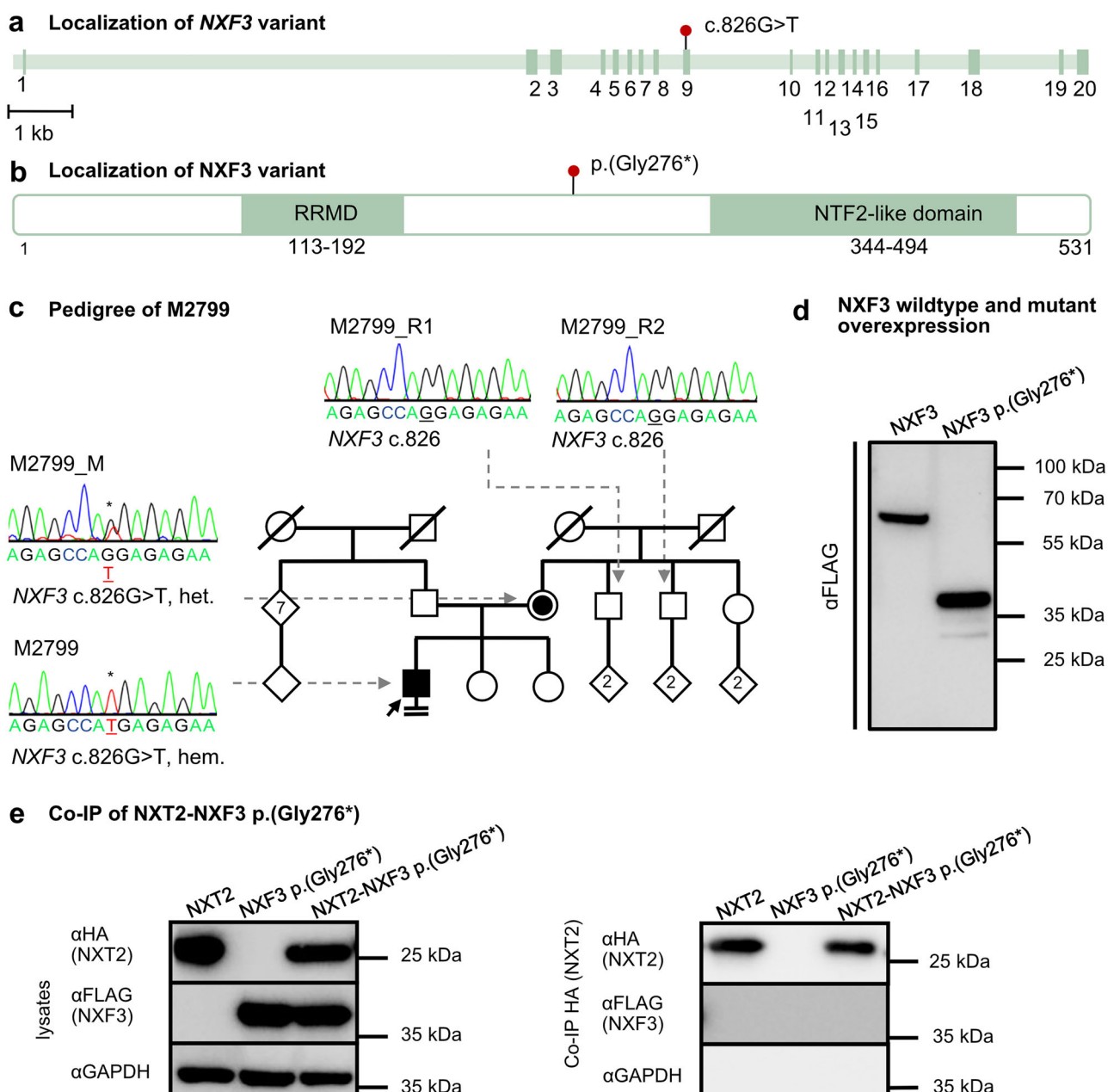

**Fig. 6 | Identification of a hemizygous stop-gain variant in *NXF3* in a proband with severely reduced sperm count. a** Schematic depiction of the exon-intron-structure of *NXF3* with the variant c.826G>T in exon nine indicated by a red dot. **b** At the protein level, the variant leads to the inclusion of a premature stop codon p.(Gly276*), resulting in a truncated protein lacking the NTF2-like domain if the mutant transcript is not degraded by NMD. **c** Pedigree of M2799 demonstrating hemizygous presence of (c).826G>T in the index case. The variant is inherited from the heterozygous mother and two fertile brothers of the mother are negative for the variant. **d** In vitro, the variant leads to the expression of a C-terminally truncated NXF3-FLAG protein (expected size: -32 kDa) (*N* = 3). **e** Western blot analysis of lysates derived from overexpressed mutant NXF3 in HEK293T cells (left). Co-IP of HA-tagged NXT2 and mutant FLAG-tagged NXF3 demonstrates abolished binding of the truncated NXF3 protein to NXT2 (right) (*N* = 3). Source data are provided as a Source Data file.

identified in females, are present in the newest gnomAD population database of >800,000 individuals (v4.1.0), indicating intolerance to LoF variants, selective pressure, and/or incompatibility with heritability. From a clinical perspective, *NXT2* is a strong candidate gene for male infertility due to azoospermia but the identification of additional patients harboring variants in *NXT2* is necessary to firmly establish a clinically valid gene-disease relationship.

Since germ cells are largely absent in affected men's testes, we propose that NXT2 has a critical function either in somatic Sertoli cells, being crucial for orchestrating spermatogenesis by maintaining the spermatogonial stem cell niche and providing essential growth factors[24], and/or in first steps of germ cell development (Supplementary Fig. 12), as evidenced by the enriched expression of *NXT2* in fetal male germ cells.

In contrast to NXT1, NXT2 has undergone adaptive selection during evolution[15] and displays a testis-enriched expression only in primates, whereas the protein is ubiquitously expressed in rodents. These different evolutionary fates already point towards a tissue-specific novel function of NXT2 restricted to the primate testis. This is supported by our finding that the absence of NXT2 in humans is

## a  Papanicolaou staining of sperm from M2799

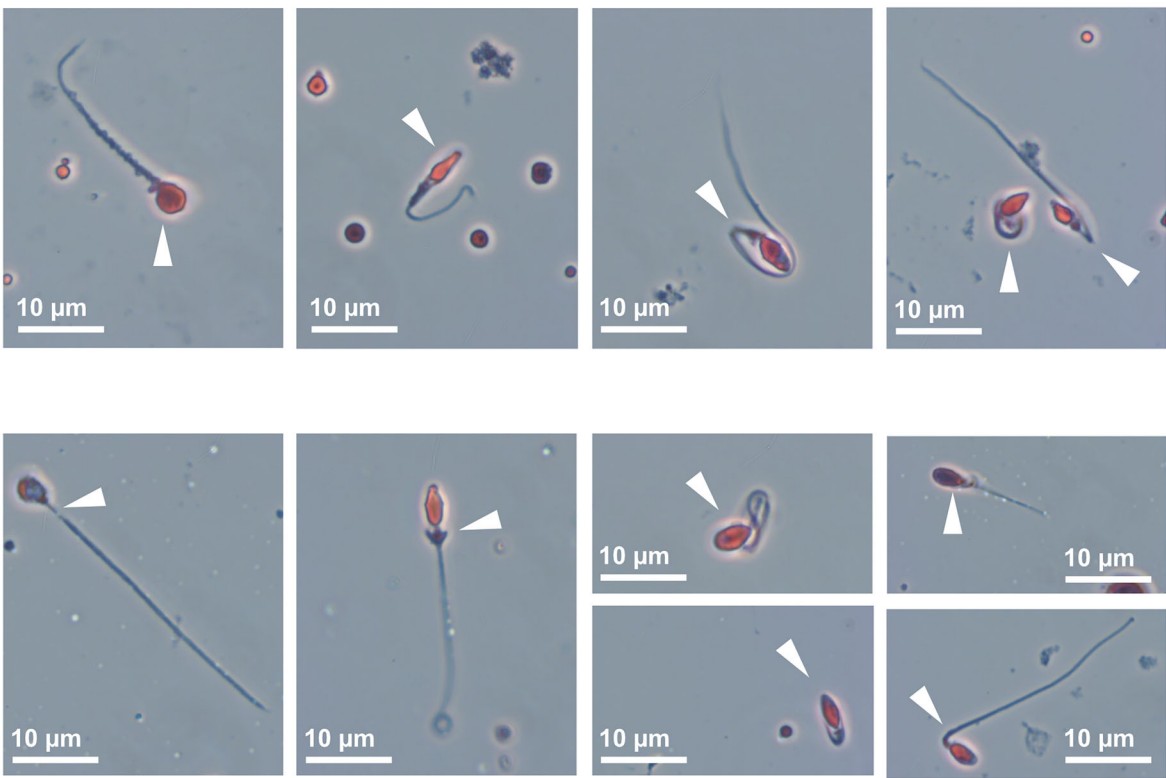

## b  NXF3, Tubulin and DAPI expression in Ctrl. and M2799 sperm

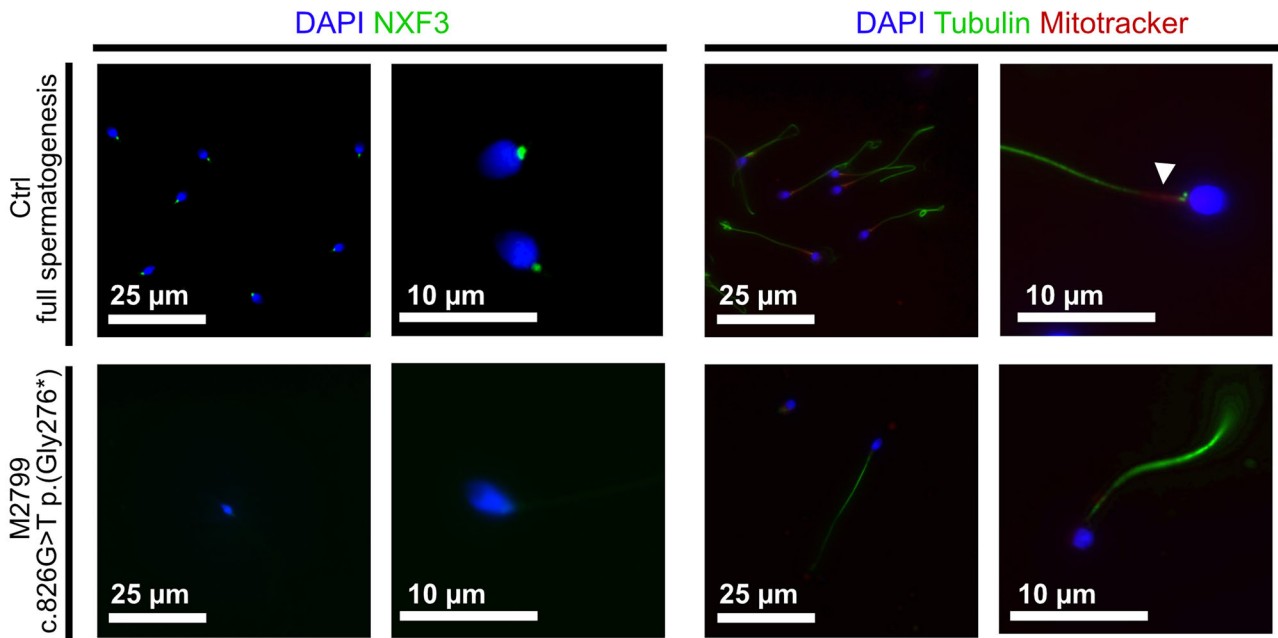

**Fig. 7 | The hemizygous LoF variant in M2799 is associated with diverse forms of structural sperm abnormalities and absence of NXF3 in the sperm midpiece.**
**a** Diverse structural defects are observed in Papanicolaou stained sperm of M2799. Abnormal sperm shapes included amorphous or round heads partially without acrosome, thin or bent midpiece, and short, bent, or tightly curled tails (*N* = 3). **b** Immunofluorescence staining for NXF3 (green) and sperm tail and midpiece marker in sperm from healthy donors and M2799. In control sperm (top left), NXF3 staining localized to the neck, very closely attached to the sperm head. In the subject's sperm (bottom left), NXF3 staining was absent, indicating absence of the protein. Sperm were characterized by staining for specific marker proteins. The head (DAPI) is stained blue, the flagellum (α/β-tubulin) is stained green and the midpiece (Mitotracker, exemplary indicated by a white arrow) is stained red (right). No sperm were found in M2799 that showed specific staining for the midpiece marker Mitotracker (right bottom) (*N* = 2).

associated with impaired spermatogenesis, whereas in mice *Nxt2* has been demonstrated to be dispensable for fertility[25].

Our data also support an involvement of NXT2 in the export of bulk mRNA through the nuclear pore, as we demonstrate not only an interaction with NXF1, part of the well-characterized NXT1-NXF1 export factor complex[5], but also with diverse nucleoporins, including NUP93 and NUP214. NUP214 contains phenylalanine-glycine (FG)-repeats that are essential for binding to NXF proteins[26] and both NUPs are involved in the transport of RNPs through the nuclear pore[27]. We also provide evidence that NXT2 is the predominant NXT protein in the human testis. Although NXT1 is capable of binding to NXF3 in vitro, it is absent from the NXF3 interactome in the human testis. Furthermore, pulldown of NXT1 from testicular lysates was not successful. These data suggest that NXT2 largely replaces the function of NXT1 in testicular nuclear export of RNA. This is further supported by the severe reproductive phenotype observed in men with *NXT2* LoF variants, indicating that the cellular function of NXT2 in the testis cannot be compensated by NXT1 in humans.

In addition to the ubiquitously expressed *NXF1*, humans encode four additional NXF genes (*NXF2-NXF5*), all of which are located on the X chromosome and display mainly tissue-specific expression patterns[8]. NXF2 shares an overall similar protein structure with NXF1, reveals a testis-specific expression, and associates with mRNA in vitro[8]. The mouse ortholog of NXF2 localizes to the nucleus or the nuclear periphery of germ cells, supporting a function also for NXF2 in RNA export through the nuclear pore complex[28]. *Nxf2* knockout mice revealed an age-dependent depletion of spermatogonial stem cells and male infertility[29], and a crucial function in regulating germ cell development was proposed. However, as no men harboring potentially deleterious variants in *NXF2* have been identified yet, it remains to be seen whether loss of NXF2 will also results in human male infertility.

The reproductive phenotypes of *NXT2* and *NXF3* LoF variant carriers revealed striking differences. While the absence of NXT2 leads to a loss of germ cells in the adult testis, spermatogenesis in the *NXF3* variant carrier proceeds to the development of structurally impaired sperm, linking the function of NXF3 to later stages of spermatogenesis compared with NXT2 (Supplementary Fig. 12). In line with this, *NXF3* is not expressed in fetal male germ cells but in post-meiotic, haploid germ cells[23], suggesting that NXF3 is dispensable for fetal functions of the NXT2-mediated cellular processes but is likely crucial for spermiogenesis, i.e. the formation of mature sperm from haploid round spermatids. Notably, and similar to *Nxt2*, knockout of *Nxf3* in mice does not impact spermatogenesis[13], again pointing towards different evolutionary fates of the human and mouse genes. Mammalian NXF3 lost its ability to bind to the nuclear pore complex[8] but gained an additional binding site for the exportin XPO1 (also called CRM1)[16]. Accordingly, based on the tissue-specific expression of *NXF3* in the testis, it has been speculated that the NXF3-XPO1 complex mediates the transport of specific RNAs[16]. We present proteomic data indicating that NXT2 is the predominant NXT protein interacting with NXF3 in the human testis. The cargoes of NXF3-mediated transport in mammals remain to be elucidated and both XPO1 and nuclear pore complex proteins were not enriched in the NXF3 interactome in this study, whereas enrichment of several proteins involved in mRNA maturation was observed. Hence, it remains to be clarified whether the infertility in the *NXF3* LoF subject is a consequence of impaired mRNA export at later stages of spermatogenesis or a consequence of other NXF3-related functions.

Examples of other RNA nuclear export-independent functions of NXF proteins have already been described for mammalian NXF2, which is involved in cytoplasmic mRNA dynamics by interacting with motor proteins such as KIF17[30]. In addition, expression of more than one NXF protein is also known from other eukaryotic lineages, including *Caenorhabditis elegans* and *Drosophila melanogaster*, and interestingly this diversification occurred independently assuming that these NXF variants might also have evolved novel molecular functions, not directly related to mRNA export[31] as we suggest for human NXF2/3. Indeed, in *Drosophila melanogaster*, one of the NXF proteins (*Drosophila* Nxf2) triggers co-transcriptional repression of transposons in germ cells[32–34]. Further, the *Drosophila* NXF3 was shown to export piRNA precursors and guide them to the nuage, a germ cell specific granule, where they are processed into mature piRNAs, which are important for protecting genome integrity by silencing transposable elements[35–37].

In summary, we introduce *NXT2* as a strong candidate gene for male infertility and demonstrate that the encoded protein is one key player in adult human testis bulk RNA nucleocytoplasmic transport by interacting with the RNA export factor NXF1 and proteins of the nuclear pore complex. NXT2 also interacts with the human testis-specific NXF1 paralogues NXF2 and NXF3 that might be involved in cellular functions independent of nucleocytoplasmic mRNA transport or developed cargo specificity in the human testis.

## Methods

### Ethical approval

All persons included in the study gave written informed consent for the analysis of their donated material and the evaluation of their clinical data compliant with local requirements. The use of testicular tissue for pulldown analysis and the MERGE study protocol were approved by the Münster Ethics Committees/Institutional Review Boards (Ref. No. Münster: 2012-555-f-S and 2010-578-f-S). Semen samples were provided by normozoospermic donors with prior written consent according to the protocols approved by the Ethics Committee of the Ärztekammer Westfalen-Lippe and the Medical Faculty Münster (4INie, 2021-402-f-S). The study protocol of the Radboudumc outpatient clinic and the Newcastle upon Tyne Hospitals NHS Foundation Trust (Newcastle, UK) was approved by the respective Ethics Committees/Institutional Review Boards (Nijmegen: NL50495.091.14 version 5.0, Newcastle: REC ref. 18./NE/0089). All procedures were in accordance with the Helsinki Declaration of 1975.

### Lysis of human testicular tissue

Adult human testicular tissue derived from surgery of three subjects with obstructive azoospermia (OA) (pooled) and two transgender person, who still had full spermatogenesis in testicular biopsy despite of the hormonal treatment, was stored at −80 °C prior homogenization with the TissueLyser LT (QIAGEN, Hilden, Germany) for 6 min with a 5 mm Stainless Steel Bead (QIAGEN, Hilden, Germany) in Pierce IP lysisbuffer (#87787 [Thermo Scientific™, Waltham, USA]; 50 mM Tris-HCl, 1% Triton X-100, 5 mM EDTA, 150 mM NaCl, 0.2 mM PMSF, 1 mM DTT, 1x protease inhibitor cocktail]. The samples were incubated on ice for 15 min and centrifuged at 4 °C and 14000 rcf for 10 min. The supernatant was immediately used for pulldown experiments.

### Pulldown of native proteins from testicular lysates and preparation of samples for mass spectrometry

Dynabeads Protein-G (Thermo Scientific™, Waltham, USA) were incubated with 5 μg of specific NXT2 and IgG antibodies (Supplementary Table 1) for 90 min at 4 °C. To perform the pulldown, the coupled beads were incubated with testicular lysate (~80 mg) at 4 °C overnight. Of the 120 μl sample, 20 μl were used for subsequent Western blot analysis to check for efficiency of the pulldown and 50 μl were washed with 50 mM Tris-HCl (pH 8) three times. The beads were dried and stored at −80 °C. Independent replicates pulled down with NXT2 (3 replicates from from testis lysate 1 & 2 each, 1 sample of pooled tissue lysates from 3 men with obstructive azoospermia, OA [HP:0011962]), NXF3 (three replicates from testis lysate 1), NXT1 (three replicates from testis lysate 1) and three (testis lysate 1) or 2 (testis lysate 2) control replicates pulled down with isotype specific IgGs were sent for mass spectrometry analyses.

## LC–ESI-MS/MS analysis

The LC–ESI-MS/MS analyses were performed in the Proteomics Facility at the University of Turku on a nanoflow HPLC system (Easy-nLC1200, Thermo Scientific™, Waltham, USA) coupled to the Q Exactive HF (Thermo Fisher Scientific, Bremen, Germany) equipped with a nano-electrospray ionization source. Peptides were first loaded on a trapping column and subsequently separated inline on a 15 cm C18 column (75 µm x 15 cm, ReproSil-Pur 3 µm 120 Å C18-AQ, Dr Maisch HPLC GmbH, Ammerbuch-Entringen, Germany). The mobile phase consisted of water with 0.1% formic acid (solvent A) and acetonitrile/water (80:20 (v/v)) with 0.1% formic acid (solvent B). A 70 min gradient was used to eluate peptides (60 min from 6% to 39 min solvent B and in 2 min from 39% to 100% of solvent B, followed by 8 min wash stage with solvent B). MS data was acquired automatically by using Thermo Xcalibur 4.1 software (Thermo Scientific™, Waltham, USA). An data dependent acquisition method repeated cycles of one MS1 scan covering a range of 350–1750 m/z followed by HCD fragment ion scans (MS2 scans) for the 10 most intense peptide ions from the MS1 scan. Data files were searched for protein identification using Proteome Discoverer 3.1 software (Thermo Fisher Scientific) connected to an in-house server running the Mascot software. Data was searched against a SwissProt Homo sapiens (2024_3) database. Protein "abundance" values, which are acquired by label-free quantification (LFQ) done by Proteome Discoverer software during the data analysis, were used to compare the amount of detected peptides between the protein-specific pull-down and the pulldown from IgG.

## Gene Ontology analysis

Gene ontology analysis (http://geneontology.org)[38,39] was performed for all genes encoding proteins significantly enriched in the NXT2 pulldown compared to IgG (Fig. 1a) (for "biological processes" and "homo sapiens") and processed with PANTHER[40] (annotation dataset: "GO biological processes complete", test type: 'Fisher's Exact', Correction: "Bonferroni", showing results with $P < 0.05$). GO terms were then processed with Revigo62 (http://revigo.irb.hr/)[41] using the $P$-value and a medium (0.7) list setting (obsolete GO terms were removed, species "homo sapiens", "SimRel" semantic similarity measure). The Revigo Table was exported and -log10($P$-value) of representative GO terms (classed as representation: 'null') were plotted with CirGO.py63[42] for visualization of the 2-tiered hierarchy of GO-terms.

## STRING interaction analysis

To identify protein clusters within the significantly enriched proteins, a STRING functional protein association analysis was carried out[43]. Analysis focused on high confidence and experimental sources. Line thickness indicates the confidence of the interaction. Proteins without any interaction with other queried proteins were not depicted. K-mean clustering with a given number of three clusters was executed.

## Cloning of cDNA constructs

Human adult testis RNA (BioCat, Heidelberg, Germany) was converted to cDNA using the GoScript™ Reverse Transcriptase system (Promega, Madison, USA) according to the instructions of the manufacturer. Amplification of human cDNAs encompassing the entire open reading frame of *NXT2* (NM_018698.5), *NXF2* (NM_022053.4) *NXF3* (NM_022052.2) and *NXT1* (NM_013248.3) was performed with PrimeSTAR Max polymerase (Takara Bio, Kusatsu, Japan) and PCR products were cloned into the mammalian expression vector pcDNA3.1(+) (Genscript, Leiden, NL). Using phosphorylated primers, an HA-tag was cloned to the N-terminus of *NXT2* and a FLAG-tag was added to the C-terminus of *NXF2* and N-terminus of *NXF3*, respectively. Deletions of respective protein domains and addition of Kozac-sequences to the 5′-end of *NXF2* and *NXF3* constructs were introduced to increase expression by PCR using phosphorylated primers and blunt end ligation of PCR products. All constructs were

verified by Sanger sequencing. Primer sequences can be found in Supplementary Table 5.

## Mutagenesis of constructs

Point mutations identified in infertile men were introduced using site-directed-mutagenesis according to manufacturer's instructions (QuikChange II XL Site-Directed Mutagenesis Kit, Agilent Technologies, Santa Clara, USA). The *NXT2* LoF variant (c.354dup), the *NXT2* missense variant (c.268 G > T) and the *NXF3* LoF variant (c.826 G > T) were introduced in wildtype (WT) *NXT2* and WT *NXF3* cDNA clones, respectively. Successful mutagenesis was verified by Sanger sequencing (see Supplementary Table 5 for primer sequences).

## Culture and transfection of HEK293T cells

Human embryonic kidney (HEK) 293T cells (Lenti-X, Clontech Laboratories, catalog number: 632180) were cultured in Dulbecco's Modified Eagle Medium (DMEM, Sigma-Aldrich, Munich, Germany), supplemented with 10% fetal calf serum (FCS) and 1% penicillin-streptomycin. The cells were maintained in T75 cell culture flasks at 37 °C and 5% $CO_2$. Passaging was accomplished twice a week, and cells were used up to passage 20. Per well 400,000 cells were seeded into 6-well plates and transfected the following day using K2 transfection reagent (Biontex Laboratories GmbH, München, Germany) with either 2 µg (single plasmid transfections) or 4 µg plasmid DNA (co-transfections). For co-transfections of two constructs, the amounts of transfected DNA were harmonized according to the length of the fragments on cDNA level. A medium change was performed 6 h after transfection and protein lysates were prepared 48 h after transfection. Transfection and subsequent analyses were performed in independent triplicates.

## Lysis of HEK293T cells

Transfected HEK293T cells were detached using ice-cold phosphate buffered saline (PBS), followed by 5 min of centrifugation at 4 °C and 2000 rcf. Lysis was performed by thorough manual pipetting. Different lysis buffers were used for standard lysates (25 mM HEPES, 100 mM NaCl, 1 mM $CaCl_2$, 1 mM $MgCl_2$, 1% TritonX-100, 1x protease inhibitor cocktail) and Co-IP samples (0.025 M Tris, 0.15 M NaCl, 0.001 M EDTA, 1% NP-40, 1% glycerol, 1x protease inhibitor cocktail). After an incubation of 15 min on ice and centrifugation for 15 min at 4 °C and 13000 rcf supernatants were separated. Samples were either directly denatured and used for Western blot or directly taken for Co-IP experiments.

## Co-IP of overexpressed proteins

HA-coupled magnetic beads (Thermo Scientific™, Waltham, USA) were used for Co-IP experiments according to manufacturer's instructions. The IP of HEK293T cell lysates was carried out 30 min at room-temperature in a rotator, followed by three (NXT2-NXF3) or nine (NXT2-NXF2) washes with 0.05% TBS-T and a final wash with $H_2O$. An acidic elution (elution buffer pH 2.0) was performed for 8 min prior pH neutralization. For each approach both co-expressed proteins were also separately transfected and used as positive and negative controls.

## Western blot

65 µl lysate were mixed with 25 µl 4x Laemmli (Bio-Rad, Hercules, USA) and 10 µl DTT and denatured at 95 °C for 10 min. Samples were separated on Mini-PROTEAN® TGX StainFree™ Precast gels (Bio-Rad, Hercules, USA) and transferred to a PVDF membrane using a Trans-blot Turbo Mini Transfer Pack kit (Bio-Rad, Hercules, USA) according to manufacturer's instructions. Membranes were blocked with 5% milk powder in 0.025% TBS-Tween (TBS-T) solution for 30 min at room temperature (RT) prior to primary antibody incubation (Supplementary Table 1) overnight at 4 °C. After washing, a peroxidase-conjugated secondary antibody incubation (2 h, RT, Supplementary Table 1) followed. Chemiluminescence was detected with the Clarity™ Western

ECL Substrate kit (Bio-Rad, Hercules, USA) and the ChemiDoc MP Imaging System (BioRad, Hercules, USA). To assess and confirm the molecular weights of analyzed proteins, a PageRuler™ plus prestained protein ladder (Thermo Scientific, Waltham, USA) was used.

## Study cohorts

The Male Reproductive Genomics (MERGE) cohort included data of 2,703 men (2,629 with exome and 74 with genome sequencing, mean age: 34) mainly recruited in the Centre of Reproductive Medicine and Andrology (CeRA) in Münster with various infertility phenotypes. All probands underwent routine semen analysis according to the WHO guidelines.

Most men of this cohort had azoospermia ($N = 1,622$, HP:0000027) or severely reduced sperm counts: $N = 487$ with cryptozoospermia (sperm only identified after centrifugation of the ejaculate, HP:0030974); $N = 168$ with extreme oligozoospermia (total sperm count <2 million, HP:0034815); $N = 85$ with severe oligozoospermia (sperm count <10 million, HP:0034818). History of oncologic diseases, including testicular tumors, as well as numerical chromosomal aberrations, such as Klinefelter syndrome (karyotype 47,XXY) and Y-chromosomal AZF-deletions, led to an exclusion. Likely pathogenic monogenic causes for the infertility phenotype have already been described in about 8% of cases[18]

One further subject with a deletion of the entire *NXT2* gene was identified among the cohort of 677 infertile men from Nijmegen/Newcastle and the respective variant has already been mentioned in the Supplementary data in a recent publication[19].

## Exome sequencing, variant filtering, and validation of sequence variants

Sequencing for the MERGE cohort has been described previously[44]. In brief, genomic DNA was extracted from peripheral blood leukocytes via standard methods. For exome sequencing of the MERGE cohort, the samples were prepared, and enrichment was carried out according to the protocol of either Agilent's SureSelectQXT Target Enrichment for Illumina Multiplexed Sequencing Featuring Transposase-Based Library Prep Technology (Agilent) or Twist Bioscience's Twist Human Core Exome. To capture libraries, Agilent's SureSelect Human All Exon Kits V4, V5 and V6 or Twist Bioscience's Human Core Exome plus RefSeq spike-in and Exome 2.0 plus comprehensive spike-in were used. For whole genome sequencing of samples from the MERGE cohort, sequencing libraries were prepared according to Illumina's DNA PCR-Free library kit. For multiplexed sequencing, the libraries were index tagged using appropriate pairs of index primers. Quantity and quality of the libraries were determined with the ThermoFisher Qubit, the Agilent TapeStation 2200, and Tecan Infinite 200 Pro Microplate reader, respectively. Sequencing was performed on the Illumina NextSeq 500 System, the Illumina NextSeq 550 System, or the NovaSeq 6000 System, using the NextSeq 500/550 V2 High-Output Kit (300 cycles), or the NovaSeq 6000 S1 and S2 Reagent kits v1.5 (200 cycles), respectively. After trimming of remaining adapter sequences and primers with Cutadapt v1.15[45], reads were aligned against GRCh37.p13 using BWA Mem v0.7.17[46]. Base quality recalibration and variant calling were performed using the GATK toolkit v3.8[47] with haplotype caller according to the best practice recommendations. For more recent samples and whole genome samples Illumina Dragen Bio-IT platform v4.2 was used for alignment and variant calling. Both pipelines use GRCh37.7.p13 as reference genome. Resulting variants were annotated with Ensembl Variant Effect Predictor[48]. Exome/genome data were screened for rare (minor allele frequency [MAF] ≤ 0.001 in gnomAD v2.1.1) variants in *NXT2*, *NXF1*, *NXF2*, and *NXF3* with a predicted effect on protein function, including copy number variants, loss-of-function variants (frameshift, stop-gain, start-loss, splice site) and amino acid substitutions with a CADD score ≥ 10.

For the Nijmegen/Newcastle cohort, blood samples from probands and saliva samples from parents were used to extract DNA using the QIAGEN® Gentra® Puregene® DNA extraction kit according to manufacturer's instructions (QIAGEN®, Venlo, NL). Samples were prepared and enriched according to manufacturer's protocols of either Illumina's Nextera DNA Exome Capture kit or Twist Bioscience's Twist Human Core Exome Kit for exome sequencing, samples submitted for whole genome sequencing were prepared instead following the manufacturer's instructions for the Ilumina TruSeq DNA PCR-free® library preparation kit followed. All samples were sequenced on a NovaSeq 6000 Sequencing System (Illumina, San Diego, USA). Sequenced reads were aligned to the Genome Reference Consortium human assembly 38 (GRCh38/hg38) through BWA-MEM. Single nucleotide variations and small indels were identified and quality-filtered using GATK's HaplotypeCaller v4.2.6.1[49]. Copy number variation (CNV) analysis of whole genome sequencing data was performed by combining Dysgu-sv[50] and GATK-based CNVRobot (https://github.com/AnetaMikulasova/CNVRobot) with default parameters. Afterwards, CNVs predicted to be de novo were inspected through IGV[51].

All identified variants were verified by Sanger sequencing; for primer sequences, see Supplementary Table 5. If available, segregation analyses were carried out with DNA from family members. PCR products were purified and sequenced using standard protocols.

## Minigene assay

To assess the impact on splicing of the missense variant c.268 G > T p.(Ala90Ser) that affects the first nucleotide of exon 4 in *NXT2*, an in vitro splicing assay based on a minigene construct was performed. Primers flanking exon 4 of *NXT2* (Supplementary Table 5) were used to amplify the region of interest from genomic DNA of M2004 as well as a human control sample by standard PCR using 0.4 U of Phusion™ High-Fidelity DNA Polymerase (Thermo Scientific™, Waltham, USA). PCR products were cloned into pENTR™/D-TOPO® (Thermo Scientific™, Waltham, USA) according to manufacturer's instructions. Gateway cloning was performed using Gateway™ LR Clonase™ Enzyme Mix (Thermo Scientific™, Waltham, USA) and pDESTsplice as destination vector (pDESTsplice was a gift from Stefan Stamm (Addgene plasmid #32484))[52]. A transient transfection with X-tremeGENE™ 9 transfection reagent (Sigma-Aldrich, St. Louis, USA) of Human Embryonic Kidney cells, HEK293T293T Lenti-X (Clontech Laboratories, Inc., Mountain View, USA) was carried out with mutant and wildtype *NXT2* minigenes. 24 h after transfection, total RNA was extracted using the RNeasy Plus Mini Kit (QIAGEN, Hilden, Germany) and reverse-transcribed into cDNA with the ProtoScript II First Strand cDNA Synthesis Kit (New England Biolabs GmbH, Frankfurt am Main, Germany). Amplification of the region of interest was performed using primers annealing to the rat insulin exons 3 and 4 that are part of the minigene construct (Supplementary Table 5). PCR products were separated on a 2% agarose gel, bands were extracted using the QIAquick Gel Extraction Kit (QIAGEN, Hilden, Germany) and sequenced (Supplementary Table 5). For better visualization, PCR products were additionally analyzed on a 4150 TapeStation System (Agilent Technologies, Santa Clara, USA).

## Period acid-Schiff staining of human testicular tissue

Subjects M2004, M3065 and RU00584 underwent testicular biopsy, as indicated by diagnosis of non-obstructive azoospermia (NOA) according to EAU guidelines[53,54] with the aim of testicular sperm extraction. After written informed consent, testicular biopsies were taken, immediately fixed in Bouin's solution, and subsequently embedded in paraffin wax. Periodic acid-Schiff (PAS) staining was carried out according to previously published protocols[55,56]. In brief, sections were dewaxed in solvent (ProTaqs Clear, #4003011; Quartett Immunodiagnostika and Biotechnologie, Berlin, Germany), rehydrated in a decreasing ethanol series and then incubated for 15 min in 1%

periodic acid. After washing with $dH_2O$ sections were incubated for 45 min with Schiff's reagent (Roth, Karlsruhe, Germany). Histological evaluation was performed following score count analysis[55]. Images were taken using an Olympus BX41 bright-field microscope equipped with a Leica DMC4500 camera.

## Histological staining of NXT2, NXT1, SOX9, MAGEA4, DDX4 and SMA in control and subjects' testicular tissue

For immunohistochemistry (IHC), deparaffinization was carried out using NeoClear (Merck, Darmstadt, Germany) and rehydration was done in a graded ethanol-water series. Antigen retrieval was performed at 90 °C in citrate buffer (pH 6) for 10 min. After 30 min of pre-blocking with 25% goat serum, the primary antibody (Supplementary Table 1) was applied and incubated in humidity chambers at 4 °C overnight. The secondary, biotinylated α-rabbit and α-mouse (ab6012; ab5886; Supplementary Table 1) antibodies were applied and incubated for 1 h at RT. Visualization was conducted using the avidin-biotin complex method for 45 min at RT (ABC solution−horseradish peroxidase (Vector Laboratories, Burlingame, USA)) followed by 3,3′-diamino-benzidine as substrate and counterstaining with hematoxylin. As a negative control, isotype rabbit IgG was included instead of the primary antibody in corresponding concentrations and an omission control was performed only using 5% BSA/TBS instead of the primary antibody. Images were obtained using the PreciPoint O8 scanning microscope system.

## Histological analysis of subject's sperm

For histological analysis and immunofluorescence staining of patient's sperm cover slides were coated with poly-L-lysine. For controls, each cover slide was incubated with ~2,000,000 sperm from a normo-zoospermic donor after swim-up as previously described[57]. As M2799's sperm are immotile, no swim-up was performed but the sperm were washed and concentrated by centrifugation before adding to the cover slide. Fixation was done using 4% PFA for 15 min.

Sperm morphology was evaluated using Papanicolaou staining. Cover slides were immersed in 96% ethanol, 80% ethanol, 70% ethanol, 50% ethanol and $dH_2O$ for 30 s, respectively. Next, the sperm were stained with hematoxylin staining solution for 5 min and washed with water for 1 min. Subsequently, the sperm was differentiated in 0.2% hydrochloric acid and washed in running water for 10 min. Then, the cover slides were immersed in 50% ethanol, 70% ethanol, 80% and 96% ethanol for 30 s. Next, sperm were stained with orange G staining solution for 1,5 min and immersed in 96% ethanol for 30 s, respectively. Afterward, sperm were stained with Polychrom solution for 1,5 min. Last, the cover slides were immersed in 96% ethanol and 99% ethanol for 30 s, respectively and immersed in rotisol for 10 min. Images of unstained sperm were obtained by DIC microscopy using the Olympus IX73. Images of stained sperm were done using the Zeiss AXIO Lab.A1.

For immunofluorescence staining cover slides with fixed patient's sperm and control sperm were washed with 0.1% Triton-X in PBS, cells were blocked with 1% BSA in 0.1% Triton-X in PBS. Primary antibody incubation was done overnight in a humidity chamber at 4 °C with specific antibodies against NXF3 and Tubulin (Supplementary Fig. 1c, d, Supplementary Table 1). After washing, DAPI (#D1306 (Thermo Scientific™, Waltham, USA)) and secondary antibodies (Supplementary Table 1) were combined for 1 h incubation at RT (protected from light). In case of additional staining with midpiece marker Mitotracker Red CMXRos (#9082 (Cell Signaling Technology, Danver, USA), 500 nM), Mitotracker Red CMXRos was incubated for 1 h at RT after the secondary antibody and DAPI were washed away with PBS-Triton-X. After mounting with Dako Fluorescence Mounting Medium (#S3023 (Dako Denmark, Glostrup, Denmark)), slides were dried at RT overnight and then stored at 4 °C prior imaging. Imaging was done using the Leica DM2500 microscope and the Leica K3M camera.

## Prediction of NXT2 3D structure from AlphaFold2

3D structures of NXT2 were predicted and edited using AlphaFold2[58,59]. The short isoform of NXT2 (NM_001242617.2) is depicted, as it is the only available version at the last date of accession (04/24/24). Of note, for all other experiments and figures, the longer isoform of NXT2 (NM_018698.5) is used as the first exon which is only present in this isoform (NM_018698.5) is testis expressed according to GTEx[60] (obtained from the GTEx Portal on 03/21/24).

## Statistics and reproducibility

In the mass spectrometry data analysis, statistical comparisons between IgG and NXT2 sample groups were performed by one-sided unpaired (in case the sample and control group refer to identical amounts of samples) or heteroscedastic $t$-test (in case sample and control group refer to different amount of samples). Experimental replicates were performed as indicated in the respective figure legends. All reported variants were validated by Sanger sequencing. The investigators were not blinded to allocation during experiments and outcome assessment.

## Reporting summary

Further information on research design is available in the Nature Portfolio Reporting Summary linked to this article.

## Data availability

Submission of human exome/genome sequencing data from the MERGE cohort to public databases is not covered by the probands informed consent. These data will be available upon request for academic use and within the limitations of the proband's informed consent from the Frank Tüttelmnn, Director of the Institute of Reproductive Genetics, Münster, Germany, by contacting frank.-tuettelmann@ukmuenster.de. Each request will be reviewed within 1 month and the researcher will need to sign a data access agreement. Sequencing data from the Nijmegen cohort have been deposited in the European Genome-phenome Archive (EGA) under the accession code EGAS00001005417. These data will be available upon request for academic use and within the limitations of the provided informed consent by applying for access through the EGA's online form. Every request will be reviewed by the Newcastle University Male Infertility Genomics Data Access Committee and the researcher will need to sign a data access agreement after approval. AlphaFold2 structure accession code for NXT2 is AF-Q9NPJ8-F1 [https://alphafold.ebi.ac.uk/entry/Q9NPJ8]. The mass spectrometry proteomics data have been deposited the ProteomeXchange Consortium via the PRIDE[61] partner repository with the dataset identifier PXD052904 and PXD061059. Novel genetic variants identified in *NXT2* and *NXF3* have been deposited in ClinVar with accession numbers SCV005043065 [https://www.ncbi.nlm.nih.gov/clinvar/variation/3235105], SCV005043066 [https://www.ncbi.nlm.nih.gov/clinvar/variation/3235106], SCV005043067 [https://www.ncbi.nlm.nih.gov/clinvar/variation/3235107] and VCV003602633.1 [https://www.ncbi.nlm.nih.gov/clinvar/variation/3602633]. Source data are provided with this paper.

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

## Acknowledgements

This study relied on data from probands who gave their permission for genomic analyses, and the authors gratefully thank all participants and their family members. We further would like to thank Luisa Meier and Christina Burhöi for excellent technical assistance, Vesna Bojovic for help in evaluating sperm morphology and Sandra Laurentino for critical reading of the manuscript. The study was carried out within the frame of the Deutsche Forschungsgemeinschaft (DFG, German Research Foundation) funded Clinical Research Unit 'Male Germ Cells' (CRU326, project no. 329621271, grants to F.T. and N.N.). F.T. was supported by the Interdisciplinary Centre for Clinical Research Münster (IZKF, Tüt4/011/23). S.A.K. was supported by the Medical Faculty Münster's 'CareerS' programme. N.K. was funded by the Novo Nordisk Foundation and the Jane and Aatos Erkko Foundation. The authors acknowledge Biocenter Finland and the use of the Turku Proteomics Facility's service for generation of mass.spectrometry data. Biorender (biorender.com) was used to create parts of Fig. 1c and Supplementary Fig. 12.

## Author contributions

Study conceptualization: A.-K.D., B.S., F.T. Data curation: A.-K.D., B.S. Funding acquisition: F.T. Investigation: A.-K.D., A.A., L.M., L.H., G.H., C.K., O.K., S.A.K., M.J.X., J.V., S.K., N.N., N.K., B.S., F.T. Visualization: A.-K.D., B.S. Writing of original draft: A.-K.D., B.S., F.T. All authors revised and approved the final version of the manuscript.

## Funding

## Competing interests

The authors declare no competing interests.
