## [Transparent Peer Review file · Nature Communications]

NXT2 is a key component of the RNA nuclear export factor complex in the human testis and essential for spermatogenesis

Corresponding Author: Dr Birgit Stallmeyer

Version 0:

Reviewer comments:

Reviewer #1

(Remarks to the Author)

The work by Dicke et al stems from the foundation of DNA sequencing of human infertile men diagnosed male-infertility (I think either no or very low sperm count). Based on loss of function changes in several men, NXT2 was identified as gene likely to be involved in male infertile, and present within the X-chromosome. Some of the loss of function variants led to no expression. In addition, NXF3 was also identified as a likely important for spermatogenesis and clinically relevant. This is an excellent paper where the authors have really started with a clinical observation, then used “basic” science to demonstrate the changes are likely pathogenic. I have toyed with the idea of asking them to make a complete knockout, but the weight of evidence on the involvement of NXF2/NXF3 is so high, I don't personally believe there as a need. As such, I really only have minor comments for the authors to address which is trying to tidy up the interpretation and present the work so everybody can understand it.

-Lines 70-72, please include the original citations that document this work.

-P5, Line 101 I think it might refer to Supplementary Table 2, but here there is no NUP93 listed as mentioned on P5, line 101 of the manuscript. Do you mean NUP98? Or NUP 93? (suppl. II)

-P5 line 112 (and page 6, line 124). When you say “protein structures” I immediately think you are referring to crystal structure. Could you perhaps describe the domains (i.e. all of them have an RNA recognition motif, LRR domain etc). This might be beneficial as XF3 looks more unique than the NXF1 and NXF2.

-Supplementary II is hard to understand. I would personally prefer to see a table with shows the spectral counts of the NXT2 pulldown compared to the antibody controls from the human testis lysate. What is protein “abundance” (also mentioned on P18, line 410) Is a number referring to the integrated MS1 count? Or MS2 counts? In this context it is difficult to understand what “significantly enriched” means. If you gave the spectral counts in addition to, or besides these “abundance numbers” I think most readers would understand how you are interpreting “significantly enriched”. For example, if the control “abundance” is simply baseline counts, there is no peptide to fragment and therefore a “0” would appear in the control column.

In addition (suppl II), there are two control IgG columns and 3 for NXT2 – how do you get a T-test from this?

-P6 lines 132-133. Please describe the observation. It appears to me that NXF2 lacking the RRM is “still able to bind to NXT2, albeit with reduced capacity”.

P9 lines 194-208. It would be helpful to those unfamiliar with molecular biology to point out exact location within the protein sequence the mutation occurs i.e. amino acids 64 of isoform 1, NXT2. The confusing bit I found was the ASP119 – which is referring to the open reading frame. However, I believe the typical naming convention, for example the one Gnomad would use, would be D64*. You did this for another variant (P10, line 225 “affects the alanine residue at position 90”). If you could check throughout the manuscript throughout please for this.

P9 lines 209-216 –The 42 kb deletion does this encompass any other genes on the X-chromosome?. For the sake of

transparency, I think it is worth mentioning that lncRNA are also present in this domain, and if so, any other protein expression gene(s).

P11 lines 236 onward. The title of this is “oligoasthenoteratozoospermia” and I suspect this is correct. As you appear to have access to the sperm samples, can you quantify morphology using DIC microscopy. It would be good to quantify the percentage of morphologically abnormal sperm, even if it is 100% to truly show the men is OAT.

Finally, can you make mention of frequency of the different MAF given you have a large dataset on hand. On this note, it wasn't clear if the authors have uploaded their human dataset to a public repository? They mention they have 2700 well characterized in fertile men. Is the plan to make this publicly available, or “only by request”.

Reviewer #2

(Remarks to the Author)

In their manuscript “NXT2 is the key player for nuclear RNA export in the human testis and critical for spermatogenesis” Dicke et al. report on the essentiality of human NXT2 for germ cell development in the male germline. The authors uncover the NXT2 interactome in human testis, validate some of the identified interactors and describe a series of human NXT2 and NXF3 mutants and the defects these mutants cause in testis.

We consider the evolution of specialized export pathways and NXT / NXF paralogs an interesting topic, but find the mechanistic insights obtained from the presented data to be limited, primarily because the function of NXT2 versus NXT1 is not experimentally addressed. While the manuscript is well written and easy to follow, the figures are crafted well, and existing literature is taken into consideration, we think that addressing the following comments would strengthen the manuscript:

1. What is the specific function of NXT2 versus NXT1 in human testis? The authors suggest that NXT2 in the testis functionally replaces NXT1. However, according to publicly available expression data also NXT1 is also expressed in human testis. From their IP-MS experiment the authors deduce that “NXT2 is the main interaction partner of NXF1 in human testis”. Experimentally this is not tested: because NXT1 / NXT2 binding to NXF proteins would be mutually exclusive, NXT1 could not be detected when pulling on NXT2. To test this the authors would need to do a testis IP on NXT1 or NXF1.

2. The authors claim that NXT2 is a specialized NXT protein, functionally distinct from NXT1. However, an alternative hypothesis would be that NXT1 and NXT2 are functionally equivalent, and that NXT2 is required in the testis because overall higher NXT levels are required due to the expression of the NXF2 and NXF3 paralogs. We understand that the availability of human testis for experiments is limited, but the authors could test:

- whether in a cell line an NXT1 knockout can be rescued by NXT2 expression,
- whether NXF2 and NXF3 can interact with NXT1.

3. The title of the paper states that “NXT2 is the key player for nuclear RNA export...”. This would be correct, if NXT1 would have no function in testis, which the authors did not test. We would further argue that every essential cellular pathway is “critical for spermatogenesis”. The authors should consider a title more reflective of their work.

4. The presence of alternative NXF proteins in testis suggest that multiple NXF dependent pathways might be at play. As has been previously shown in literature cited by the authors (Batki et al. 2019, also Fabry et al., 2019, eLife 8: e47999 and Zhao et al., 2019, Nat. Cell Biol. 21(10): 1261—1272) NXF1 paralogs are not necessarily involved in nuclear RNA export. The analysis in Fig. 1c-d thus seem of limited value to us, as the annotation of NXF2 and NXF3 by homology might be incorrect. This extends to Fig. 1e: the authors propose that in the testis NXT2 is the main NXF1 interactor, this is not reflected in this figure. Please adjust these figures and text statements accordingly.

Further points:

1. Line 58-59: the authors state that the NXF1 RRM is essential for binding mRNA. This claim needs an appropriate citation, we are in fact unaware of an experiment that probed this.
2. Fig. 1a / Supplemental Table S2: In Fig. 1e the authors list NUP93, which is missing from Sup. Table S2. The supplemental table instead lists NUP98, which however is enriched only 2.3 fold.
3. Fig. 1a/b: SPATA5 and SPATA5L1 are not validated. It cannot be excluded that the polyclonal NXT2 antibody used in the IP has off targets, and this unexpected result should thus be taken with great caution, in particular in the discussion (lines 349-356).
4. Fig. 1f: Expression in which tissue are the authors showing here? Would it not be more informative to show expression levels in testis here?
5. Fig. 2: The cartoon schematics showing protein domains that were deleted would be more intuitive to follow if (a) there were shown underneath each other and (b) where excised regions are shown as thin black lines.
6. Given that a broad audience is targeted, a cartoon schematic of testis development would be helpful for the relevant manuscript figures.
7. Fig. S5f: How come that the blot image for NXT2 looks spliced, while the anti-Flag and anti-GAPDH blots aren't? Are the samples not run on the same gel?

Reviewer #3

(Remarks to the Author)

I reviewed the paper by Dicke et al., titled "NXT2 is the Key Player for Nuclear RNA Export in the Human Testis and Critical for Spermatogenesis," with great interest. The study presents an important investigation into NXT2, a gene located on the X chromosome that plays a critical role in male germ cell development in humans. While NXT2 is ubiquitously expressed in mice, its expression in humans is testis-enriched, indicating an evolutionarily specialized role in spermatogenesis.

The authors used mass spectrometry following NXT2 pull-down from adult human testis tissue to identify the interactome of NXT2. Among the eight proteins identified, NXF2 and NXF3, both members of the nuclear export factor family encoded by X-chromosomal genes, were highlighted. These findings suggest that NXT2 operates in a testis-specific RNA export pathway similar to that of its paralog NXT1, which is involved in ubiquitous mRNA export. The authors conclude that NXT2 plays a central role in this pathway with NXF2 and NXF3 in testis. To further validate these interactions, the authors used a heterologous expression system followed by co-immunoprecipitation (Co-IP) experiments, demonstrating that the interaction depends on the NTF2-like domain present in all these factors. They also explored the clinical relevance of NXT2, identifying two hemizygous variants in a cohort of 2,700 infertile men. One variant was a missense mutation affecting partially splicing, but further functional tests did not confirm its pathogenicity. The second was a frameshift variant that segregated with the disease in a family. Additionally, a de novo deletion of NXT2 was identified in another cohort of 667 infertile men, with all patients displaying non-obstructive azoospermia due to Sertoli-cell-only syndrome. The study also identified a hemizygous stop-gain variant in NXF3 in one patient with severe oligoasthenozoospermia. While this variant produced a truncated protein lacking the NTF2-like domain, inconsistencies between immunofluorescence and western blotting results require further investigation. The lack of mitochondria in the sperm cells of this patient remains unexplained, and additional functional studies on NXF3 are needed to better understand its role.

Overall, this study is well-conducted, with an original approach. The results are well-presented, thoroughly discussed, and the manuscript is clear and easy to follow. However, further clarification of the NXF3-associated phenotypes and its role in spermatogenesis would strengthen the study. That said, such experiments may require considerable time and effort, and could form the basis for a separate study focused specifically on the function of this factor and its involvement in male infertility.

I recommend the manuscript for publication in its current form.

Version 1:

Reviewer comments:

Reviewer #1

(Remarks to the Author)

The authors have been fair with my comments and concerns raised and I am of the opinion they have answered them respectively and judiciously.

Reviewer #2

(Remarks to the Author)

In the revised version, the authors have mostly addressed our comments, yet the following points remain unclear and warrant rephrasing:

1. The authors state that "the non-canonical RNA-recognition motif domain (RRMD) has been shown to contribute to the minimal RNA binding domain (PMID: 10202158)" to confirm that NXF1 binds mRNA through this domain. Yet, unless we misread the paper, this is not stated in the referenced publication. The referenced study assessed that the NXF1 RRM domain contributes to binding of the viral CTE element, which however is thought to not reflect how mRNA engages with NXF1 (PMC3167930), which is to our knowledge still unknown. The cited paper does not support the authors claim and this claim should either be substantiated with another experiment, an appropriate reference, or corrected.

2. Line 102: The authors write: "[...] it is likely that the encoded protein [NXT2] has evolved to acquire functions and binding partners distinct from its paralog NXT1." What is the evidence that supports this strong claim? A claim of novel functions requires either experimental support or should be rephrased. This is not apparent to us from the data provided. If anything, the provided data suggest that NXT2 is functionally equivalent to NXT1 in testis tissue.

3. Page 9, Line 175: "Interestingly, NXT1 was absent from the NXT2-NXF interactome, suggesting that it cannot compensate for the testicular function of NXT2." This statement does not make sense to us. If NXT1 is mutually exclusive with binding to NXT2, then it could not IP with NXT2. This result does not suggest that it cannot compensate for NXT2 function. Please clarify. The related comment in Line 384 does similarly not make sense: "no NXT1 was observed in the NXT2 interactome, even though the well-known interaction partner NXF1 was enriched." NXT1 could not be observed in the NXT2 interactome, if it binds mutually exclusively with NXT1 to NXF proteins. Please rephrase.

Point by point response to the reviewers' comments

Title: NXT2 is a key component of the RNA nuclear export factor complex in the human testis and essential for spermatogenesis

Manuscript number: NCOMMS-24-47023

Revision version: 1

Editor's decision received date: 2024-11-13

Dear editors and reviewers,

Thank you very much for reviewing our manuscript, we sincerely appreciate the overall positive comments on our study and your helpful feedback. We have critically revised the manuscript according to your suggestions and respond to them point by point below.

All line numbers refer to the version with tracked changes.

Reviewer #1 (Remarks to the Author):

The work by Dicke et al stems from the foundation of DNA sequencing of human infertile men diagnosed male-infertility (I think either no or very low sperm count). Based on loss of function changes in several men, NXT2 was identified as gene likely to be involved in male infertile, and present within the X-chromosome. Some of the loss of function variants led to no expression. In addition, NXF3 was also identified as a likely important for spermatogenesis and clinically relevant. This is an excellent paper where the authors have really started with a clinical observation, then used "basic" science to demonstrate the changes are likely pathogenic. I have toyed with the idea of asking them to make a complete knockout, but the weight of evidence on the involvement of NXF2/NXF3 is so high, I don't personally believe there as a need. As such, I really only have minor comments for the authors to address which is trying to tidy up the interpretation and present the work so everybody can understand it.

Dear reviewer,

Thank you very much for your positive review of our manuscript. We are very pleased that you evaluate our manuscript as "excellent". Please find below our responses to your very valuable comments.

-Lines 70-72, please include the original citations that document this work.

Thank you for highlighting this point. We rephrased the sentence and added the original citations (lines 72-75).

-P5, Line 101 I think it might refer to Supplementary Table 2, but here there is no NUP93 listed as mentioned on P5, line 101 of the manuscript. Do you mean NUP98? Or NUP 93? (suppl. II)

Thank for spotting this mistake. NUP93 is correct, and NUP98 in the Supplementary Table should have been NUP93. Since we have added data on additional NXT2 pulldowns in the manuscript, the data presented in the former Supplementary Table 2 are now shown in more detailed Supplementary Data 1-5 in Excel format.

-P5 line 112 (and page 6, line 124). When you say “protein structures” I immediately think you are referring to crystal structure. Could you perhaps describe the domains (i.e. all of them have an RNA recognition motif, LRR domain etc). This might be beneficial as XF3 looks more unique than the NXF1 and NXF2.

Thank you for this suggestion. We changed the wording from “protein structure” to “protein sequence” and included a description of the shared and unique domains of the different NXF proteins. To avoid repetitions, we moved this paragraph to the section “NXT2 interacts with NXF2 and NXF3 through NTF2-like domains” (lines 150-155).

-Supplementary II is hard to understand. I would personally prefer to see a table with shows the spectral counts of the NXT2 pulldown compared to the antibody controls from the human testis lysate. What is protein “abundance” (also mentioned on P18, line 410) Is a number referring to the integrated MS1 count? Or MS2 counts? In this context it is difficult to understand what “significantly enriched” means. If you gave the spectral counts in addition to, or besides these “abundance numbers” I think most readers would understand how you are interpreting “significantly enriched”. For example, if the control “abundance” is simply baseline counts, there is no peptide to fragment and therefore a “0” would appear in the control column. In addition (supp II), there are two control IgG columns and 3 for NXT2 – how do you get a T-test from this?

Thank you for pointing this out. We used the “abundance values” for presenting results and calculating statistics from the mass spectrometry data because it is a more accurate method than spectral counting. The abundance values are acquired by label-free quantification (LFQ) done by Proteome Discoverer software during the data analysis. LFQ is a MS1-based quantification and it is a measurement of the precursor ion intensity (e. g. peak height at the apex of the chromatographic profile for each peptide). The analysis software Proteome Discoverer determines these intensity values by determining the maximum peptide spectrum match height of the extracted ion chromatogram. It then sets this intensity as the intensity of the peptide group to which the peptide spectrum match belongs. Protein intensities are calculated as a sum of the top 3 most intense peptide group areas associated with that protein. Spectral counting is an earlier approach for label-free quantification which has been largely

replaced by precursor intensity measurements. To make this clearer to the reader we included a more detailed description of the mass spectrometry data analysis in the Method section and clearly describe what data the abundance values represent (lines 507-513).

Concerning your question on the significance, we have changed the focus for the presentation of the mass spectrometry data by sorting the proteins according to their x-fold enrichment, rather than on a significant p-value. Additionally, we added data on a new NXT2 pulldown performed on 3 samples each for NXT2 and IgG control. The testicular lysates used for these experiments were obtained from a different donor than in the initial pulldown. These data are now presented in the Results (lines 102-119) and also shown in new Supplementary Data 1 (raw data) and 2 (proteins sorted according to their x-fold enrichment compared to IgG control) and confirm the main results of the initial pulldown experiment. The data from the initial NXT2 pulldown are presented in Supplementary Data 3-5. For the new pulldown experiment we performed a one-sided unpaired T-Test because the control and sample groups had identical sizes (data are shown in Supplementary Data 2). Since the sample sizes differed between control group and NXT2 group in the NXT2 pulldown from lysate 2 (initial pulldown experiment) we performed a one-sided heteroscedastic T-Test (Supplementary Data 4). We added this information to the Method section (lines 734-738).

-P6 lines 132-133. Please describe the observation. It appears to me that NXF2 lacking the RRM is “still able to bind to NXT2, albeit with reduced capacity”.

Thank you for raising this point. We agree that in the blot you highlight there is still a faint signal detectable in the NXT2-NXF2^{NTF2^{Del}} Co-IP Western blot. However, this signal was also visible when the Co-IP was performed in the absence of NXT2 and we concluded that it is due to an unspecific binding of the FLAG-tagged NXF2^{NTF2^{Del}} protein to the beads. We now increased the number of washing steps during the co-immunoprecipitation, and in the new α -FLAG Western blots, there's no signal visible (see Figure 2b).

P9 lines 194-208. It would be helpful to those unfamiliar with molecular biology to point out exact location within the protein sequence the mutation occurs i.e. amino acids 64 of isoform 1, NXT2. The confusing bit I found was the ASP119 – which is referring to the open reading frame. However, I believe the typical naming convention, for example the one gnomAD would use, would be D64*. You did this for another variant (P10, line 225 “affects the alanine residue at position 90”). If you could check throughout the manuscript throughout please for this.

Thank you for your comment. ASP119 refers to the long isoform of NXT2 (NM018698.5) (isoform 1), which has also been used for cloning the constructs for Co-IP analysis. D64 refers to NXT2 isoform NM_001242617.2, which encodes a smaller NXT2 that lacks 55 N-terminal amino acids. To make this clearer, we included the NM number and the ENST number of the NXT2 and NXF3 isoforms the variant descriptions are referring to in the text and describe the

position in more detail (lines 220-222, lines 313-315). The identifiers can also be used to find the data for the respective variant in gnomAD [<https://gnomad.broadinstitute.org/transcript/ENST00000372103>].

P9 lines 209-216 –The 42 kb deletion does this encompass any other genes on the X-chromosome?. For the sake of transparency, I think it is worth mentioning that lncRNA are also present in this domain, and if so, any other protein expression gene(s).

Thank you for this valuable comment. The deleted region does not contain any additional protein or long-coding RNA encoding genes but it covers an enhancer of *KCNE5*. We added this information to the results (lines 278-282).

P11 lines 236 onward. The title of this is “oligoasthenoteratozoospermia” and I suspect this is correct. As you appear to have access to the sperm samples, can you quantify morphology using DIC microscopy. It would be good to quantify the percentage of morphologically abnormal sperm, even if it is 100% to truly show the men is OAT.

Thank you for highlighting this topic. We re-analysed the sperm morphology of M2799 by DIC and also performed Papanicolaou staining on sperm samples. Images of abnormal sperm shapes identified are now shown in a new Figure 7a and a new Supplementary Figure 10. The percentages of morphological abnormal sperm affecting the sperm head, the midpiece and the sperm tail are now given in the text (lines 330-332).

Finally, can you make mention of frequency of the different MAF given you have a large dataset on hand. On this note, it wasn't clear if the authors have uploaded their human dataset to a public repository? They mention they have 2700 well characterized in fertile men. Is the plan to make this publicly available, or “only by request”.

Similar to other genetic variants causing male infertility, the minor allele frequencies of the variants reported here in the disease population are quite low. In case of the MERGE cohort 2,362 exomes of men with diverse subforms of reduced sperm count have been analyzed (numbers are given in the Methods, lines 597-600), which would refer to a MAF of 0.0004 for each of the single nucleotide variants identified. However, since we believe that much larger cohorts are needed to calculate meaningful data on disease-specific MAFs we prefer not to add this information to the manuscript. The exome sequencing data from the Nijmegen cohort have been submitted to EGA. Submission of human exome/genome sequencing data from the MERGE cohort is not covered by the probands informed consent. These data will be made available upon request for academic use and within the limitations of the proband's informed consent from the corresponding author. This is now indicated in the data availability statement (lines 743-756).

Reviewer #2 (Remarks to the Author):

In their manuscript “NXT2 is the key player for nuclear RNA export in the human testis and critical for spermatogenesis” Dicke et al. report on the essentiality of human NXT2 for germ cell development in the male germline. The authors uncover the NXT2 interactome in human testis, validate some of the identified interactors and describe a series of human NXT2 and NXF3 mutants and the defects these mutants cause in testis.

We consider the evolution of specialized export pathways and NXT / NXF paralogs and interesting topic, but find the mechanistic insights obtained from the presented data to be limited, primarily because the function of NXT2 versus NXT1 is not experimentally addressed. While the manuscript is well written and easy to follow, the figures are crafted well, and existing literature is taken into consideration, we think that addressing the following comments would strengthen the manuscript:

Dear reviewer,

Thank you very much for evaluating our manuscript and your helpful comments. Please find below our answers to your suggestions that definitely helped to improve the manuscript.

1. What is the specific function of NXT2 versus NXT1 in human testis? The authors suggest that NXT2 in the testis functionally replaces NXT1. However, according to publicly available expression data also NXT1 is also expressed in human testis. From their IP-MS experiment the authors deduce that “NXT2 is the main interaction partner of NXF1 in human testis”. Experimentally this is not tested: because NXT1 / NXT2 binding to NXF proteins would be mutually exclusive, NXT1 could not be detected when pulling on NXT2. To test this the authors would need to do a testis IP on NXT1 or NXF1.

Thank you for highlighting this important point. We agree that according to publicly available transcriptomics data, *NXT1* mRNA is present in the human adult testis, and we have included this information now also in the manuscript (lines 177-178, Figure 1d). Following your suggestions, we not only repeated the pulldown of NXT2 from testicular tissue of a second human donor but also performed a pulldown of NXT1 and NXF3 from the identical testicular lysates. Similar to NXT2, validated antibodies that were shown to specifically recognizing recombinant NXT1 or NXF3 proteins were used for these pulldown experiments (Supplementary Figure 1). The new NXT2 pulldown confirmed the main results of the data we showed in the first submission (Supplementary Data 1&2; Figure 1a&b). Additionally, we now show that only NXT2 is enriched in the NXF3 pulldown, while NXT1 is absent in all three replicates (Supplementary Data 6-8). Furthermore, NXT1 was also absent in the mass spec data of the NXT1 pulldown (Supplementary Data 9), although we used extreme high

concentrations of the validated monoclonal antibody, specifically recognizing NXT1 but not NXT2. Accordingly, we not only describe the results of the NXT2 pulldown in the results section in more detail (lines 99-124) but also included a novel paragraph “NXT2 is the predominant NXT protein in the human adult testis and part of the NXF3 interactome” (lines 173-205).

2. The authors claim that NXT2 is a specialized NXT protein, functionally distinct from NXT1. However, an alternative hypothesis would be that NXT1 and NXT2 are functionally equivalent, and that NXT2 is required in the testis because overall higher NXT levels are required due to the expression of the NXF2 and NXF3 paralogs. We understand that the availability of human testis for experiments is limited, but the authors could test:

- whether in a cell line an NXT1 knockout can be rescued by NXT2 expression,
- whether NXF2 and NXF3 can interact with NXT1.

Thank you for your suggestions. Since we are focusing on the testicular function of NXT2, the rescue of the function of a NXT1 knockout in a cell line derived from other human tissues would not be informative for the obviously special situation in the human testis. Unfortunately, for germ cells no cell culture model is available, so we could not perform rescue experiments for a knockout of NXT2 by NXT1. However, according to your suggestion we analyzed the possible interaction of NXT1 with NXF2 and NXF3 by Co-IP (new Supplementary Figure 7). Not unexpectedly, NXT1 is able to bind to NXF2 and NXF3 *in vitro*, arguing that the absence of NXT1 in the NXF3 pulldown is not related to the inability of both proteins to interact but more likely related to low NXT1 protein expression in the human testis. This finding is supported by the NXT1 pulldown results (Supplementary Data 9, Supplementary Figure 2c). In addition, we performed immunohistochemical staining for NXT1 in sections of human testicular tissue with complete spermatogenesis using three different NXT1 antibodies (including the validated NXT1-specific antibody used for pulldown, which was shown to specifically detect recombinant NXT1 overexpressed in HEK cells). For none of these antibodies, a specific staining pattern could be identified. We show the IHC result for the validated NXT1 antibody in a novel Supplementary Figure 8. We therefore highlight that NXT1 and NXT2 are functional equivalent at least with respect to their ability to bind to NXF proteins, but NXT2 is the predominant NXT protein in the human adult testis (lines 173-205).

3. The title of the paper states that “NXT2 is the key player for nuclear RNA export...”. This would be correct, if NXT1 would have no function in testis, which the authors did not test. We would further argue that every essential cellular pathway is “critical for spermatogenesis”. The authors should consider a title more reflective of their work.

Thank you for this comment. We believe that our genetic data clearly demonstrate that loss of NXT2 is linked to human spermatogenic failure, a phenotype that is in first line related to impairment of germ cell specific pathways as meiosis, piRNA related function or impaired

hormone signaling but not a necessary consequence of any essential cellular pathway. In addition, we now show that NXT2 is the main NXT protein in the human testis. However, since we do not prove that the NXT2-NXF complex binds to mRNA we changed the title to: “NXT2 is a key component of the RNA nuclear export factor complex in the human testis and essential for spermatogenesis”

4. The presence of alternative NXF proteins in testis suggest that multiple NXF dependent pathways might be at play. As has been previously shown in literature cited by the authors (Batki et al. 2019, also Fabry et al., 2019, eLife 8: e47999 and Zhao et al., 2019, Nat. Cell Biol. 21(10): 1261—1272) NXF1 paralogs are not necessarily involved in nuclear RNA export. The analysis in Fig. 1c-d thus seem of limited value to us, as the annotation of NXF2 and NXF3 by homology might be incorrect. This extends to Fig. 1e: the authors propose that in the testis NXT2 is the main NXF1 interactor, this is not reflected in this figure. Please adjust these figures and text statements accordingly.

Thank you for highlighting this point. We agree that for NXF1 paralogs in different species, cellular functions distinct from RNA nuclear export have been described. Accordingly, we mention this in the Discussion, taking into account also the two additional citations you highlight. (lines 432-443). However, human NXF3 has been demonstrated to be essential for mRNA export (PMID: 11545741), although this is independent from a direct binding to NUP proteins. In addition, also NXF1 and several nuclear pore complex proteins are present in the NXT2 interactome. We therefore still believe that the gene ontology analysis reflects not only results based on protein homology to NXT1 and NXF1. To address the points you raised, we moved the results of the GO analysis to the Supplementary Information (Supplementary Figure 3) and adjusted Figure 1e (now Figure 1c).

Further points:

1. Line 58-59: the authors state that the NXF1 RRM is essential for binding mRNA. This claim needs an appropriate citation, we are in fact unaware of an experiment that probed this.

Thank you very much for addressing this inaccuracy. Indeed, the non-canonical RNA-recognition motif domain (RRMD) has been shown to contribute to the minimal RNA binding domain (PMID: 10202158), which has also been shown for the four leucine rich repeats (PMID: 10202158). In addition, also the transport factor 2 (NTF2)-like domain was described to be important for cargo mRNA binding (PMID:25628355). Accordingly, we adapted the wording in the respective paragraph and added appropriate citations (lines 58-65).

2. Fig. 1a / Supplemental Table S2: In Fig. 1e the authors list NUP93, which is missing from Sup. Table S2. The supplemental table instead lists NUP98, which however is enriched only 2.3 fold.

Thank you for spotting this inaccuracy, which was also highlighted by Reviewer 1. NUP93 would have been correct. Since we have added data on additional NXT2 pulldowns in the manuscript the data presented in the former Supplementary Table 2 are now shown in a more detailed Supplementary Table 3 in Excel format.

3. Fig. 1a/b: SPATA5 and SPATA5L1 are not validated. It cannot be excluded that the polyclonal NXT2 antibody used in the IP has off targets, and this unexpected result should thus be taken with great caution, in particular in the discussion (lines 349-356).

Taking into account your feedback we deleted the respective sentences in the discussion (lines 444-451).

4. Fig. 1f: Expression in which tissue are the authors showing here? Would it not be more informative to show expression levels in testis here?

In this Figure we depict the mRNA expression in the human testis in comparison to the expression in other tissues to show that expression of NXT2, NXF2 and NXF3 is enriched in the human testis, while NXT1 and NXF1 are ubiquitously expressed. The red dots in the Figure 1d corresponds to the expression of the respective gene in the human testis, which is now indicated in the Figure legend. We changed the respective Figure slightly (Figure 1d) to make it easier to compare the expression levels between the autosomal genes NXT1 and NXF1 and the X-chromosomal, testis-enriched paralogs.

Fig. 2: The cartoon schematics showing protein domains that were deleted would be more intuitive to follow if (a) there were shown underneath each other and (b) where excised regions are shown as thin black lines.

We agree and adapted the Figure accordingly.

6. Given that a broad audience is targeted, a cartoon schematic of testis development would be helpful for the relevant manuscript figures.

We included a new Supplementary Figure 12, schematically presenting the different steps of spermatogenesis and the temporal expression profiles of the different NXT1 and NXF1 paralogs and refer to this Figure in the Discussion (line 369).

7. Fig. S5f: How come that the blot image for NXT2 looks spliced, while the anti-Flag and anti-GAPDH blots aren't? Are the samples not run on the same gel?

For each of the Co-IP analysis we always loaded the identical lysates on 3 different gels and probed one gel with α HA antibody, one with α FLAG and the third with α GAPDH (Source data).

In the former Figure S5f other samples were loaded between the samples of the NXF2 and NXT2 p.(Ala90Ser) pulldown. We now repeated some of the Co-IPs and always load the samples for each blot in the identical order to avoid unnecessary cutting of blot images. The former Supplementary Figure 5 is now Supplementary Figure 9.

Reviewer #3 (Remarks to the Author):

I reviewed the paper by Dicke et al., titled “NXT2 is the Key Player for Nuclear RNA Export in the Human Testis and Critical for Spermatogenesis,” with great interest. The study presents an important investigation into NXT2, a gene located on the X chromosome that plays a critical role in male germ cell development in humans. While NXT2 is ubiquitously expressed in mice, its expression in humans is testis-enriched, indicating an evolutionarily specialized role in spermatogenesis.

The authors used mass spectrometry following NXT2 pull-down from adult human testis tissue to identify the interactome of NXT2. Among the eight proteins identified, NXF2 and NXF3, both members of the nuclear export factor family encoded by X-chromosomal genes, were highlighted. These findings suggest that NXT2 operates in a testis-specific RNA export pathway similar to that of its paralog NXT1, which is involved in ubiquitous mRNA export. The authors conclude that NXT2 plays a central role in this pathway with NXF2 and NXF3 in testis. To further validate these interactions, the authors used a heterologous expression system followed by co-immunoprecipitation (Co-IP) experiments, demonstrating that the interaction depends on the NTF2-like domain present in all these factors. They also explored the clinical relevance of NXT2, identifying two hemizygous variants in a cohort of 2,700 infertile men. One variant was a missense mutation affecting partially splicing, but further functional tests did not confirm its pathogenicity. The second was a frameshift variant that segregated with the disease in a family. Additionally, a de novo deletion of NXT2 was identified in another cohort of 667 infertile men, with all patients displaying non-obstructive azoospermia due to Sertoli-cell-only syndrome. The study also identified a hemizygous stop-gain variant in NXF3 in one patient with severe oligoasthenozoospermia. While this variant produced a truncated protein lacking the NTF2-like domain, inconsistencies between immunofluorescence and western blotting results require further investigation. The lack of mitochondria in the sperm cells of this patient remains unexplained, and additional functional studies on NXF3 are needed to better understand its role.

Overall, this study is well-conducted, with an original approach. The results are well-presented, thoroughly discussed, and the manuscript is clear and easy to follow. However, further clarification of the NXF3-associated phenotypes and its role in spermatogenesis would strengthen the study. That said, such experiments may require considerable time and effort,

and could form the basis for a separate study focused specifically on the function of this factor and its involvement in male infertility.

I recommend the manuscript for publication in its current form.

Dear reviewer,

Thank you very much for the very positive evaluation of our study. We are delighted that you think that our study is well-conducted and shows important investigations on NXT2.

Even though you recommended the manuscript for publication in its current form, we took into account your comment on a further investigation of the NXF3-associated phenotype and present now a detailed description of the NXF3 related sperm phenotypes as obtained from DIC microscopy and Papanicolaou staining (Figure 7a, Supplementary Figure 10), demonstrating that structural impairment of patient's sperm does not only affect the midpiece but also the sperm tail and head.

Point by point response to the comments of Reviewer 2

Title: NXT2 is a key component of the RNA nuclear export factor complex in the human testis and essential for spermatogenesis

Manuscript number: NCOMMS-24-47023

Revision version: 2

Editor's decision received date: 2025-04-18

Dear Reviewer,

Thank you very much for the positive evaluation of the revised version of our manuscript. We have critically revised the manuscript according to your suggestions and respond to them point by point below.

All line numbers refer to the version with tracked changes.

Reviewer #2 (Remarks to the Author):

In the revised version, the authors have mostly addressed our comments, yet the following points remain unclear and warrant rephrasing:

1. The authors state that “the non-canonical RNA-recognition motif domain (RRMD) has been shown to contribute to the minimal RNA binding domain (PMID: 10202158)” to confirm that NXF1 binds mRNA through this domain. Yet, unless we misread the paper, this is not stated in the referenced publication. The referenced study assessed that the NXF1 RRM domain contributes to binding of the viral CTE element, which however is thought to not reflect how mRNA engages with NXF1 (PMC3167930), which is to our knowledge still unknown. The cited paper does not support the authors claim and this claim should either be substantiated with another experiment, an appropriate reference, or corrected.

Thank you very much for highlighting this inaccuracy. We corrected the sentence to: “The RRMD and the LRR both belong to the minimal RNA binding domain, that was shown to bind to the constitutive transport element (CTE) of simian type D retrovirus RNA” (lines 60-61).

2. Line 102: The authors write : “[...] it is likely that the encoded protein [NXT2] has evolved to acquire functions and binding partners distinct from its paralog NXT1.” What is the evidence that supports this strong claim? A claim of novel functions requires either experimental support or should be rephrased. This is not apparent to us from the data provided. If anything, the provided data suggest that NXT2 is functionally equivalent to NXT1 in testis tissue.

Thank you very much for this comment. We deleted this statement accordingly and rephrased the sentence to: “As *NXT2* shows a testis-enriched expression in humans and has been influenced by adaptive selection in primates¹⁵, we aimed to analyze the the interactome of *NXT2* *in vivo* by means of antibody-mediated capture of *NXT2* from testis tissue lysates.... (lines 95-99).

3. Page 9, Line 175: “Interestingly, *NXT1* was absent from the *NXT2*-*NXF* interactome, suggesting that it cannot compensate for the testicular function of *NXT2*.”. This statement does not make sense to us. If *NXT1* is mutually exclusive with binding to *NXT2*, then it could not IP with *NXT2*. This result does not suggest that it cannot compensate for *NXT2* function. Please clarify. The related comment in Line 384 does similarly not make sense: “no *NXT1* was observed in the *NXT2* interactome, even though the well-known interaction partner *NXF1* was enriched.” *NXT1* could not be observed in the *NXT2* interactome, if it binds mutually exclusively with *NXT1* to *NXF* proteins. Please rephrase.

Thank you for these suggestions. We rephrased the statement “Interestingly, *NXT1* was absent from the *NXT2*-*NXF* interactome, suggesting that it cannot compensate for the testicular function of *NXT2*” (lines 363-365) and deleted “no *NXT1* was observed in the *NXT2* interactome, even though the well-known interaction partner *NXF1* was enriched” from the respective sentence (lines 157-158).